# Transport of soluble proteins through the Golgi occurs by diffusion via continuities across cisternae

**Galina V Beznoussenko[1,2†], Seetharaman Parashuraman[2,3†], Riccardo Rizzo[3], Roman Polishchuk[2,4], Oliviano Martella[2], Daniele Di Giandomenico[2], Aurora Fusella[2], Alexander Spaar[2], Michele Sallese[2], Maria Grazia Capestrano[2], Margit Pavelka[5], Matthijn R Vos[6], Yuri GM Rikers[6], Volkhard Helms[7], Alexandre A Mironov[1]\*, Alberto Luini[2,3,4]\***

[1]Fondazione IFOM, Istituto FIRC di Oncologia Molecolare (IFOM-IEO Campus), Milan, Italy; [2]Department of Cell Biology and Oncology, Consorzio Mario Negri Sud, Santa Maria Imbaro, Italy; [3]Institute of Protein Biochemistry, Consiglio Nazionale Delle Ricerche (CNR-IBP), Naples, Italy; [4]Telethon Institute for Genetics and Medicine (TIGEM), Naples, Italy; [5]Department of Cell Biology and Ultrastructure Research, Center for Anatomy and Cell Biology, Medical University of Vienna, Vienna, Austria; [6]FEI Company, Eindhoven, Netherlands; [7]Center for Bioinformatics, Saarland University, Saarbruecken, Germany

**\*For correspondence:** alexandre.mironov@ifom.eu (AAM); luini@tigem.it (AL)

†These authors contributed equally to this work

**Competing interests:** The authors declare that no competing interests exist.

**Abstract** The mechanism of transport through the Golgi complex is not completely understood, insofar as no single transport mechanism appears to account for all of the observations. Here, we compare the transport of soluble secretory proteins (albumin and α1-antitrypsin) with that of supramolecular cargoes (e.g., procollagen) that are proposed to traverse the Golgi by compartment progression–maturation. We show that these soluble proteins traverse the Golgi much faster than procollagen while moving through the same stack. Moreover, we present kinetic and morphological observations that indicate that albumin transport occurs by diffusion via intercisternal continuities. These data provide evidence for a transport mechanism that applies to a major class of secretory proteins and indicate the co-existence of multiple intra-Golgi trafficking modes.

## Introduction

Nearly one third of the eukaryotic proteins are synthesized at the endoplasmic reticulum (ER) and then transported to their cellular destinations through the secretory pathway. Over the years, the general organization of membrane transport along the secretory pathway has been gradually unraveled (*Mellman and Simons, 1992*; *Mellman and Warren, 2000*), and many of the underlying molecular components have been identified (*Rothman, 2002*; *Schekman, 2002*; *Emr et al., 2009*). Some key questions, however, remain unresolved (*Pfeffer, 2007*; *Emr et al., 2009*; *Glick and Luini, 2011*).

A central issue is how cargo proteins traverse the Golgi complex (*Malhotra et al., 1989*; *Glick and Malhotra, 1998*; *Glick and Luini, 2011*), a major transport station composed of stacks of flat membranous cisternae. There are three main anterograde transport mechanisms that are in principle possible and might apply to the Golgi: (a) transport by compartment progression–maturation; (b) transport by dissociative anterograde vesicular carriers and, (c) transport via inter-compartment continuities.

Among these, the progression–maturation model has gained a degree of consensus as an intra-Golgi traffic mechanism, based on several lines of evidence in mammals (*Bonfanti et al., 1998*; *Lanoix et al., 2001*; *Martinez-Menarguez et al., 2001*; *Mironov et al., 2001*; *Rizzo et al., 2013*), yeast

**eLife digest** The Golgi is a structure within cells where proteins and other large molecules are modified and prepared for delivery to locations inside or outside of the cell. Each Golgi is made from a stack of flattened sacs called cisternae that are filled with fluid and enclosed by a membrane.

Proteins and other molecules are transported to the Golgi by packages called vesicles, which fuse with the outermost cisterna, which is known as the 'cis-face' of the Golgi, and unload their contents. From here, the proteins are processed and modified by enzymes as they move through the Golgi towards the 'trans-face' on the opposite side. The modified proteins are then re-packaged into vesicles before being sent to their intended destinations.

But how do proteins move through the Golgi? Some researchers have suggested that proteins do not actually move: rather, the stacks of the Golgi move like a conveyer belt as new cisterna are added to the cis-face. However, other researchers have proposed that molecules proceed from one cisterna to the next inside small vesicles. It is also possible that proteins are transported through the Golgi in other ways, or by a combination of two or more methods.

Now, Beznoussenko, Parashuraman et al. reveal that some small, soluble, proteins can move through the Golgi by diffusion. These proteins move much quicker than large protein complexes, which suggests that multiple transport mechanisms do co-exist within the Golgi. Furthermore, Beznoussenko, Parashuraman et al. found that these soluble proteins are most likely moving through some narrow tunnel-like connections between the individual cisternae.

Following on from the work of Beznoussenko, Parashuraman et al., the main challenge is to understand how all the different types of proteins that move through the Golgi are transported—which includes roughly a third of all human proteins. As many of these proteins are important for human health, learning to control their transport might create new opportunities to understand and treat disease.

(*Losev et al., 2006*; *Matsuura-Tokita et al., 2006*; *Rivera-Molina and Novick, 2009*), algae (*Becker et al., 1995*), and plants (*Donohoe et al., 2013*). Under this model, cargo molecules remain in the lumen of the Golgi cisternae while the cisternae themselves progress through the stack and 'mature' through recycling of their resident enzymes. Recently, cisternal progression has been proposed to apply only to the rims (and not to the core) of the cisternae in the mammalian Golgi (*Lavieu et al., 2013*). In addition to the Golgi, the progression–maturation principle appears to be involved in the endocytic (*Rink et al., 2005*; *Poteryaev et al., 2010*) and the phagocytic pathways (*Fairn and Grinstein, 2012*) in different species.

The vesicular transport mechanism, whereby dissociative carriers transport cargoes between successive compartments, operates at many stages of the trafficking pathway and has been proposed to apply also to intra-Golgi trafficking (*Rothman, 2002*). Here, however, the evidence is less direct and less conclusive than at other transport segments, with conflicting claims about the presence (*Orci et al., 2000*) or absence (*Claude, 1970*; *Sabesin and Frase, 1977*; *Severs and Hicks, 1979*; *Clermont et al., 1993*; *Dahan et al., 1994*; *Di Lazzaro et al., 1995*; *Bonfanti et al., 1998*; *Orci et al., 2000*; *Martinez-Menarguez et al., 2001*; *Mironov et al., 2001*; *Gilchrist et al., 2006*) of anterograde cargo proteins in the peri-Golgi carriers. Moreover in particular cases, like in microsporidia, intra-Golgi transport appears to occur without COPI vesicles (*Beznoussenko et al., 2007*).

Diffusion-based transport via inter-compartment continuities remains the least explored and understood of the traffic mechanisms. Some antecedents, however, are available. Continuity-mediated transport has been observed to occur between endosomes and lysosomes (*Luzio et al., 2007*), and also the exocytic release of cargo from secretory granules (*Rutter and Hill, 2006*) or synaptic vesicles through transient pores (kiss-and-run) (*Rizzoli and Jahn, 2007*; *Alabi and Tsien, 2013*) at the plasma membrane can be considered to occur via this modality. For intra-Golgi transport, this mechanism has been discussed several times in the past (*Mellman and Simons, 1992*; *Weidman, 1995*; *Mironov et al., 1997*; *Marsh et al., 2004*; *Trucco et al., 2004*; *Mironov et al., 2005*; *Beznoussenko et al., 2007*; *Glick and Luini, 2011*) and a few recent intra-Golgi transport models including the mixing–partitioning (*Patterson et al., 2008*), the kiss-and-run (*Mironov and Beznoussenko, 2012*; *Fusella et al., 2013*; *Mironov et al., 2013*) and the cisternal progenitor schemes (*Pfeffer, 2010*) have been

proposed that imply transient tubular continuities across cisternae. At the molecular/mechanistic level, Golgi tubule formation has been proposed to be initiated by COPI coatomer-mediated budding (*Yang et al., 2011*), and tubule elongation and fission appear to require the actions of cytosolic phospholipase A2 (cPLA2) and lysophosphatidic acid acyltransferase-γ (LPAATγ), respectively (*San Pietro et al., 2009*) (*Yang et al., 2011*). Recent evidence also points to a role for Golgi localized SNAREs and BARS in the dynamics of the intercisternal connections (*Fusella et al., 2013*). Nevertheless, a complete understanding of the molecular players regulating the intra-Golgi connections remains lacking.

Altogether, uncertainties remain about the applicability of continuity-based transport to the Golgi. One main reason for this situation has been the long-standing difficulty of demonstrating intercisternal continuities in thin sections for electron microscopy. This obstacle has now been partly overcome by the use of electron tomography and new methods of three dimensional electron microscopy (*Briggman and Bock, 2012*), which have revealed the presence of intercisternal tubular continuities under experimental conditions that favor the detection of these tubules, such as the induction of active trafficking (*Marsh et al., 2004*; *Trucco et al., 2004*; *Vivero-Salmeron et al., 2008*; *San Pietro et al., 2009*; *Wanner et al., 2013*). The second and main problem, yet to be resolved, is that the mere presence of intercisternal tubules is insufficient to prove a role for these continuities in transport, as these tubules might be too few and unfavorably disposed to support trafficking.

To test the continuity-based transport model, it is thus necessary to search for functional evidence of a transport role for these continuities. To this end, we have used soluble secretory proteins as transport markers, as these are globular objects of a few nm in diameter that should easily cross the observed intercisternal tubules and rapidly move from the cis to the trans face of an interconnected stack or ribbon. The transport of some soluble proteins has been studied decades ago using electron microscopic autoradiography (*Caro and Palade, 1964*; *Jamieson and Palade, 1967*; *Ashley and Peters, 1969*; *Castle et al., 1972*) and biochemical pulse-chase assays (*Jamieson and Palade, 1967*; *Lodish et al., 1983*), but their actual mechanism of secretion remains unknown.

Comparing the trafficking pattern of prototypic soluble proteins with those of cargoes previously proposed to move by cisternal progression–maturation, we find that soluble proteins cross the Golgi stack at a much faster rate, apparently by diffusion along intercisternal connections; and that this transport mode coexists in the same Golgi complex with the much slower intra-Golgi progression of large, non-diffusible cargo, such as procollagen I (PC-I). Soluble secreted proteins are of great physiological interest because they represent a significant portion (possibly more than 10%) of the mammalian proteome and include hormones, growth factors, serum proteins, antibodies, and digestive enzymes. Thus, these results are consistent with a novel mechanism of transport for a major class of secretory proteins, and provide evidence for multiplicity of transport mechanisms that can help to rationalize most of the observed intra-Golgi trafficking patterns.

## Results

### The experimental system: comparing transport of soluble cargo with that of VSVG and PC

As prototypes of soluble proteins we used albumin and α1-antitrypsin (hereinafter termed antitrypsin). These are globular, water-soluble proteins roughly 3 nm in diameter that should easily diffuse through the 30–60 nm wide Golgi intercisternal connections (*Trucco et al., 2004*). Albumin is an abundant, non-glycosylated protein, while antitrypsin is N-glycosylated. The trafficking of soluble proteins (albumin in most experiments) was characterized and compared with that of PC-I (*Weinstock and Leblond, 1974*; *Bonfanti et al., 1998*; *Mironov et al., 2001*) and vesicular stomatitis virus G protein (VSVG) (*Bergmann and Singer, 1983*; *Mironov et al., 2001*; *Patterson et al., 2008*), because these cargoes have been extensively characterized and shown to move by cisternal progression (or rimmal progression [*Lavieu et al., 2013*] or compartment progression [*Mironov et al., 2013*]. For the sake of brevity, from now onward we will use the term compartment progression to describe the traffic of procollagen and other similar cargo). Thus, if albumin moves by diffusion via continuities, it should exhibit transport kinetics and patterns different from VSVG and PC-I. PC-I forms large, stable, non-diffusible aggregates that cannot enter tubules or vesicles and cross the Golgi stack in a gradual fashion by compartment progression (*Bonfanti et al., 1998*; *Trucco et al., 2004*); and VSVG is a large trimeric transmembrane viral protein that shows the same trafficking pattern as PC-I, at least under certain specific conditions (see below). In this study, we only used conditions under which VSVG crosses the Golgi by compartment progression.

# Albumin crosses the Golgi stack faster than VSVG and PC: kinetic evidence for fast synchronized albumin movement across progressing cisternae

We first compared the kinetics of intra-Golgi transport of albumin with those of VSVG and PC-I in HepG2 cells, a human hepatoma cell line that secretes both albumin and antitrypsin. To assess traffic rates, we used synchronization techniques by which cargoes can be arrested in the intermediate compartment (IC), and then released, to monitor their synchronous passage through the secretory system.

To compare albumin with VSVG, HepG2 cells were infected with VSV and subjected to the following synchronization protocol (protocol 2, 'Materials and methods'): the secretory pathway was first cleared of cargo by blocking protein synthesis with cycloheximide (CHX); and then CHX was removed at 15°C. At this temperature albumin and VSVG were re-synthesized relatively efficiently, and were then transported to, and arrested in, the IC (*Mironov et al., 2001*). Finally, the 15°C transport block was removed by shifting the temperature to 32°C, to allow the synchronous passage of albumin and VSVG from the IC to and through the Golgi complex (*Mironov et al., 2001*). Notably, this protocol does not seriously overload/perturb the secretory pathway since, under similar conditions, the Golgi complex has been shown to maintain a normal structure and function (*Trucco et al., 2004*; *Mironov et al., 2001*). To monitor cargo passage, we used both immuno-electron microscopy (immuno-EM) and immuno-fluorescence.

By immuno-EM, albumin was seen at time 0 (i.e., at end of the 15°C block) in the ER and IC at similar levels, with very little in the Golgi stacks (*Figure 1A,B*, green arrowheads). An earlier study had shown that a soluble protein (soluble secretory GFP) concentrates in the IC/Golgi area at 15°C (*Blum et al., 2000*); however, no EM experiments were carried out to verify the localization. Using immuno-EM, we do not observe any such concentration of albumin in the Golgi after the 15°C block (time 0). Albumin is clearly restricted to the ER and IC, and absent from the Golgi apparatus (*Figure 1A,B,I,L*). Within 2 min of release from the 15°C block, albumin entered and filled the entire Golgi, including the trans-Golgi network (TGN), with apparently similar levels throughout (*Figure 1C,D,I*). After 5 min at 32°C, the distribution of albumin had not changed significantly (*Figure 1E,F,I*), while at 10 min, albumin was higher in the TGN than in the cis cisternae (*Figure 1G,I*). Then (by 20 min), albumin began to exit the Golgi, as indicated by its diminishing overall levels in the Golgi stack (*Figure 1H,J,L*). In sum, albumin spreads through the stack in less than 2 min, then exits the Golgi complex.

The pattern of VSVG traffic differed from that of albumin. As previously described (*Mironov et al., 2001*; *Trucco et al., 2004*), at time 0, VSVG was depleted in the ER, concentrated in the IC, and nearly absent in the Golgi stacks (*Figure 1A,B,K*). 2 min after the 15°C block release, VSVG was still mostly in IC elements adjacent to the cis-Golgi (*Figure 1C,D,K*), and at 5 min it had reached only the first cis-cisterna (*Figure 1E,F,K*). Later, VSVG gradually reached the medial and then the trans-Golgi (*Figure 1G,K*). Thus, VSVG moves gradually through the stack in over 15 min, consistent with the compartment progression trafficking mechanism, as expected under these synchronization conditions (*Mironov et al., 2001*; *Trucco et al., 2004*).

For immunofluorescence experiments (*Figure 1—figure supplement 1*), we monitored arrival of both VSVG and albumin at the cis- and trans-Golgi by determining their degree of co-localization with cis- and trans-Golgi markers (GM130 and TGN46, respectively) (*Mironov et al., 2001*; *Trucco et al., 2004*). This is feasible because cis- and trans-Golgi markers can be resolved (to a large though not complete extent) by confocal microscopy (*Shima et al., 1997*; *Trucco et al., 2004*) ('Materials and methods'). Albumin showed a diffuse ER-like distribution at time 0, with no clear Golgi staining (*Figure 1—figure supplement 1*); then, 2 min after the release of the 15°C block, albumin entered the Golgi stack and co-localized to the same extent with both GM130 and TGN46 (i.e., it reached both the cis and trans areas, *Figure 1—figure supplement 1*), while the ER was still not completely empty. After 5–10 min, albumin had completely left the ER and now localized mostly in the Golgi, where its levels declined in the cis-Golgi, while they remained high in the trans-Golgi (*Figure 1—figure supplement 1*), compatible with rapid albumin diffusion through the stack followed by concentration in the TGN. Thus, the export of albumin out of the ER was very efficient, so that by 10 min after the release of the temperature block almost all of the protein had been transported to the Golgi apparatus. Antitrypsin showed very similar distribution and trafficking patterns to albumin (*Figure 1—figure supplement 1*).

VSVG, instead, showed a punctate (IC-like) distribution at 15°C, as previously reported (*Mironov et al., 2001*; *Trucco et al., 2004*; *Figure 1—figure supplement 1*). After the release of the block, VSVG reached the cis-Golgi first (at 5 min) (*Figure 1—figure supplement 1*), and then later, with a lag

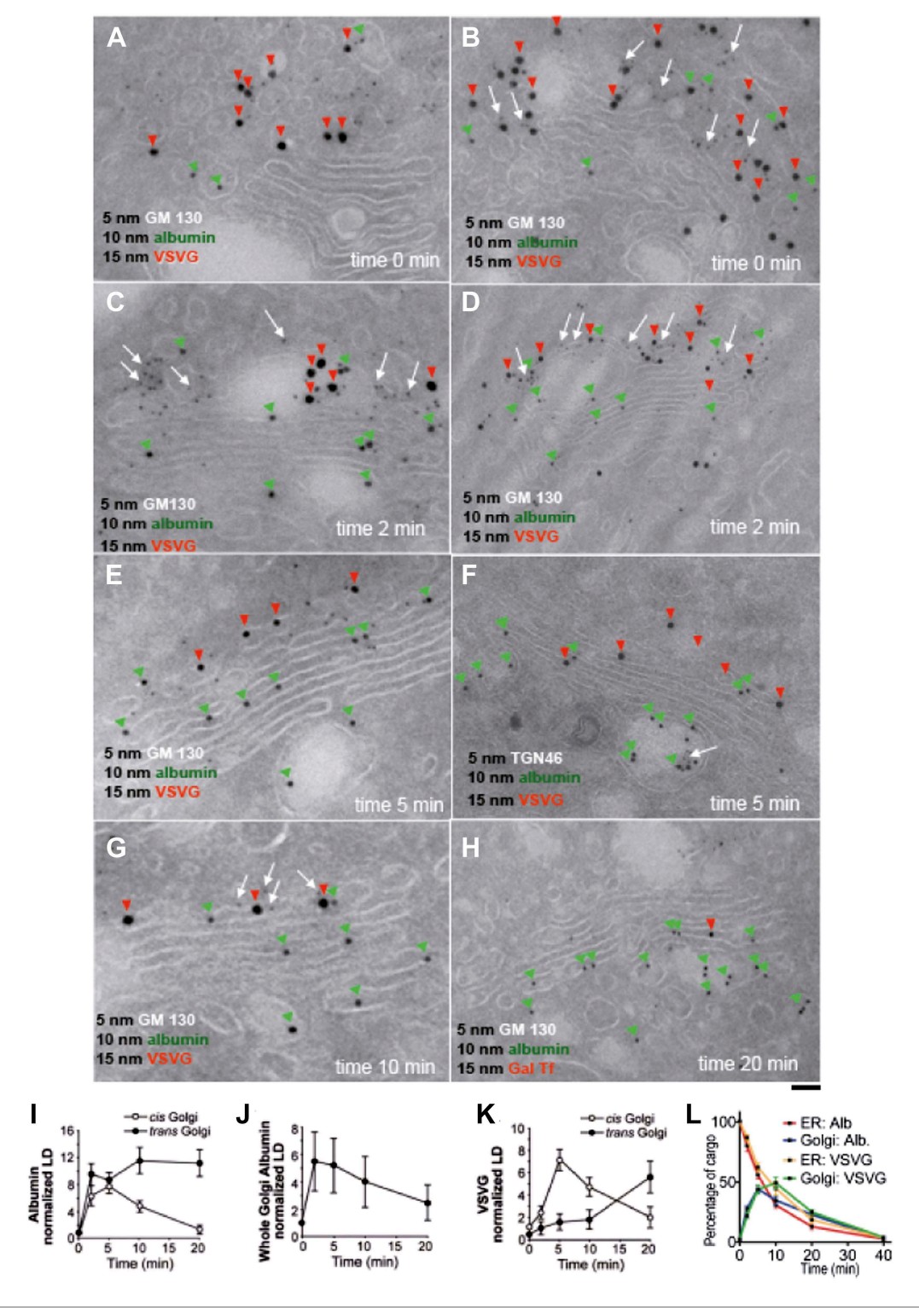

**Figure 1**. Kinetic patterns of synchronized transport of albumin and VSVG through the Golgi stack. VSV-infected HepG2 cells were synchronized according to the CHX/32-15°C protocol ('Materials and methods'). Following release of the 15°C block, the cells were examined by immuno-EM (**A–H**) at the indicated times. Panels (**I–K**) show quantification of immuno-EM values as labeling density (LD) normalized to the density in the ER, to avoid labeling variability across samples. (**L**) The amount of albumin or VSVG in indicated compartments were normalized to that
*Figure 1. Continued on next page*

*Figure 1. Continued*

present at time 0 in the ER and expressed as percentage. Values are mean ± SD from 30 stacks per time point, in three independent experiments for immuno-EM. Bar: 60 nm (**A**), 50 nm (**B**, **C**, **E**, **G**), 100 nm (**D**, **H**), 80 nm (**F**).
The following figure supplements are available for figure 1:

**Figure supplement 1**. Kinetic patterns of synchronised transport of albumin and VSVG through the Golgi stack as determined by immunofluorescence.

of 10–15 min, it arrived at the TGN, as previously described (*Mironov et al., 2001*; *Figure 1—figure supplement 1*). Again, this is compatible with compartment progression, and is in agreement with the immuno-EM data.

Next, we compared albumin and PC-I. We expressed albumin in professional PC-I secretory cells (human fibroblasts) by microinjecting albumin cDNA in the nucleus and subjecting the cells to the synchronization protocol 1 ('Materials and methods'). A limited but sufficient fraction of injected cells expressed albumin. At 15°C (time 0), albumin was mostly diffuse in the ER (as seen in HepG2 cells), while PC-I was seen in scattered fluorescent 'spots' (presumably PC-I aggregates within the IC) (*Figure 2A,B*; *Mironov et al., 2001*; *Trucco et al., 2004*). We then increased the temperature to 32°C. In these cells, the PC-I trafficking pattern has been characterized extensively in previous studies: PC-I arrives at the cis-Golgi from the IC in 2–3 min and later progresses to the TGN by compartment progression in 12–15 min (*Bonfanti et al., 1998*; *Mironov et al., 2001*). Here, we confirmed that within 3 min after the release of the 15°C block, PC-I aggregates reach the Golgi area but not the TGN (*Figure 2D,E*); and by correlative light-immuno-EM (CLEM), we further confirmed that at this time PC-I aggregates reach the cis but not the distal cisternae, well in line with previous reports (*Figure 2F*) (*Bonfanti et al., 1998*; *Mironov et al., 2001*). In the same cells, by contrast, albumin filled the Golgi stack rapidly, as in HepG2 cells: at 3 min, it already co-localized with the TGN marker TGN46 (by immunofluorescence) (*Figure 2C,E*) and, by EM, it filled the Golgi stacks from cis to trans (*Figure 2G*).

Collectively, these results indicate the existence, in the same cells (and in the same stacks), of two different intra-Golgi trafficking patterns for different cargo types, one consistent with gradual compartment progression, for PC and VSVG, and one characterized by the rapid spreading of cargo through the stack, for albumin.

## Albumin crosses the Golgi stack faster than VSVG and PC also under steady-state transport conditions

A possible limitation of these data is that they were obtained using synchronized traffic waves. Albeit relatively mild (*Mironov et al., 2001*), the traffic synchronization protocols that were applied here might 'overload' the secretory pathway. We therefore sought to examine the transport patterns of albumin and PC in cells at steady-state. This can be achieved using GFP-tagged cargoes in living HeLa cells ('Materials and methods'), which offer controlled expression conditions.

GFP-albumin showed steady-state Golgi localization and secretory behavior similar to that of native albumin in HepG2 cells (*Figure 3—figure supplement 1*), indicating that this construct can be used as an albumin tracer. Moreover, a characterization of the GFP-albumin dynamics in HeLa cells based on fluorescence recovery after photobleaching (FRAP) (*Patterson et al., 2008*), showed that this construct enters and exits the Golgi with half-times of about 3–4 min (*Figure 3—figure supplement 1*) and diffuses 'horizontally' along the Golgi ribbon in seconds, as expected from its soluble nature (*Figure 3—figure supplement 1*). We thus proceeded to assess the steady-state transport behavior of GFP-albumin, and to compare it with that of PC. To this end, we bleached the Golgi area (*Figure 3A,B*) and monitored the time for arrival of GFP-albumin from the ER at the cis-Golgi and at the trans-Golgi (again by quantifying its co-localization with GM130 and TGN46; see above and 'Materials and methods') (*Figure 3C–F*). After 1–2 min (i.e., the earliest time at which GFP-albumin had recovered to detectable levels in the Golgi stack) (*Figure 3C*), GFP-albumin had reached both the cis-Golgi and the TGN (*Figure 3D–F*); in fact, it showed a slightly higher degree of co-localization with TGN46 than with GM130 (using the unbiased co-localization Method 2 based on automatic thresholding; see 'Materials and methods'), indicating that it had already traversed the Golgi stack (*Figure 3D–F*, quantification in J). Later (3 min post-bleaching), the GFP-albumin signal became slightly

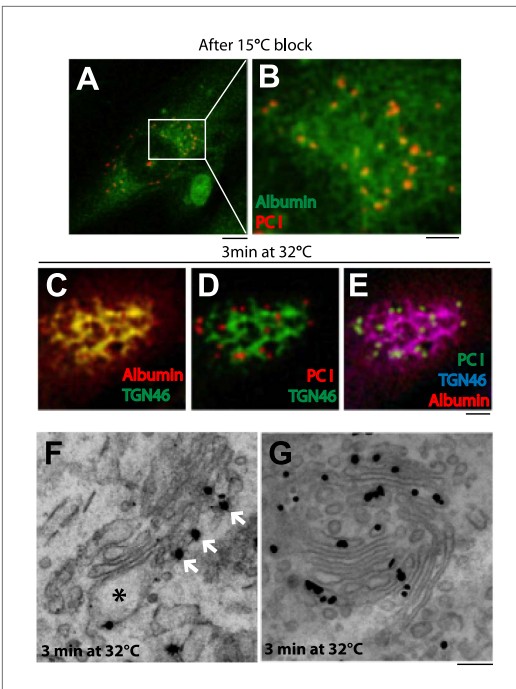

**Figure 2**. Kinetic patterns of synchronized transport of albumin and PC-I through the Golgi stack. Human fibroblast cells were microinjected in the nucleus with cDNA for albumin and incubated for 2 hr before further treatments. Transport was synchronized according to the CHX/32-15°C protocol and the cells were examined by immunofluorescence and immuno-EM. (**A**) Immunofluorescence localization of albumin and PC at the end of the 15°C block. The area in (**A**) indicated by white rectangle is enlarged in (**B**) (**C**–**E**). Co-localization between albumin (**C**) or PC (**D**) or of both cargoes (**E**) with TGN46, 3 min after release of the block. (**F**–**G**). Localization of PC (**F**) and albumin (**G**) 3 min after release of the 15°C block by immuno-EM. PC (indicated by *) localizes selectively to the cis-cisterna. The cis-side of the Golgi is revealed by the presence of GM130 labeled by immuno-nanogold technique (indicated by white arrows) (**F**). Albumin labeled by immuno-nanogold technique (black dots) shows a diffuse localization throughout the Golgi complex (**G**). Bars: 5 μm (**A**), 2 μm (**B**–**E**), 125 nm (**E** and **F**).

higher in the trans- than the cis-Golgi, and at 12 min (when recovery was nearly complete), it was clearly higher in the trans- than the cis-Golgi (as seen before bleaching, with a ratio of about 1.8) (*Figure 3J*). To control for the possibility that part of the fluorescence signal recovered in the Golgi area might come from the underlying ER, we repeated this experiment using nocodazole-induced ministacks (*Figure 3G–I,K*), where the cis and trans-Golgi markers are resolved better (*Shima et al., 1997*; *Trucco et al., 2004*) and the very low background fluorescence of the ER present in the cellular periphery allows a better resolution of Golgi fluorescence. The results were very similar to those obtained with the intact ribbon. Next, to confirm these results by EM we resorted to GFP-photooxidation followed by CLEM experiments. For photooxidation studies, the same experiments as those described above were carried out, and the cells were fixed 2 min after photobleaching, when GFP-albumin fluorescence had recovered in the Golgi. Then, the newly arrived fluorescent protein in the Golgi was excited in the presence of DAB under conditions that favor the photo-oxidation reaction and the formation of DAB electron dense precipitates in the close vicinity of GFP (*Grabenbauer et al., 2005*; *Meiblitzer-Ruppitsch et al., 2008*) ('Materials and methods'). The Golgi elements that had been monitored by video microscopy were then examined by CLEM (*Mironov and Beznoussenko, 2013* and 'Materials and methods'). The results shown in *Figure 3O–S* clearly indicated that after 2 min of recovery GFP-albumin was already filling the whole Golgi stack.

To monitor the behavior of PC under similar conditions, we used HeLa cells transfected with GFP-tagged PC-III (a homotrimer that forms large aggregates in the Golgi complex like PC-I, and in general behaves like PC-1; [*Perinetti et al., 2009*] and 'Materials and methods'). A limited but sufficient number of cells expressed this cargo. We then bleached the whole Golgi area and monitored the rate of entry of PC-III-GFP into the Golgi complex from the ER. PC-III behaved as expected from our previous experiments on PC-I trafficking (*Bonfanti et al., 1998*; *Mironov et al., 2001*; *Trucco et al., 2004*). At 3 min post-bleaching, some PC-III-GFP aggregates (in the form of distinct bright puncta) had already entered the Golgi area, but had not reached the TGN (i.e., did not co-localize with TGN46) (*Figure 3L,N*). Later (at 9 min), many more PC-III-GFP aggregates had reached the Golgi stack and some of these co-localized with TGN46 (*Figure 3M,N*), confirming that PC-III, like PC-I, enters the Golgi and then moves gradually to the TGN, consistent with compartment progression and different from the albumin transport pattern.

We also monitored whether the fast transport of soluble cargoes by diffusion is coupled to their processing by Golgi enzymes. To this end, the biochemical maturation of antitrypsin was monitored by the pulse chase assay. Antitrypsin was processed efficiently by Golgi enzymes, as evidenced by increased apparent molecular weight of the protein, with a kinetics reflecting that measured by

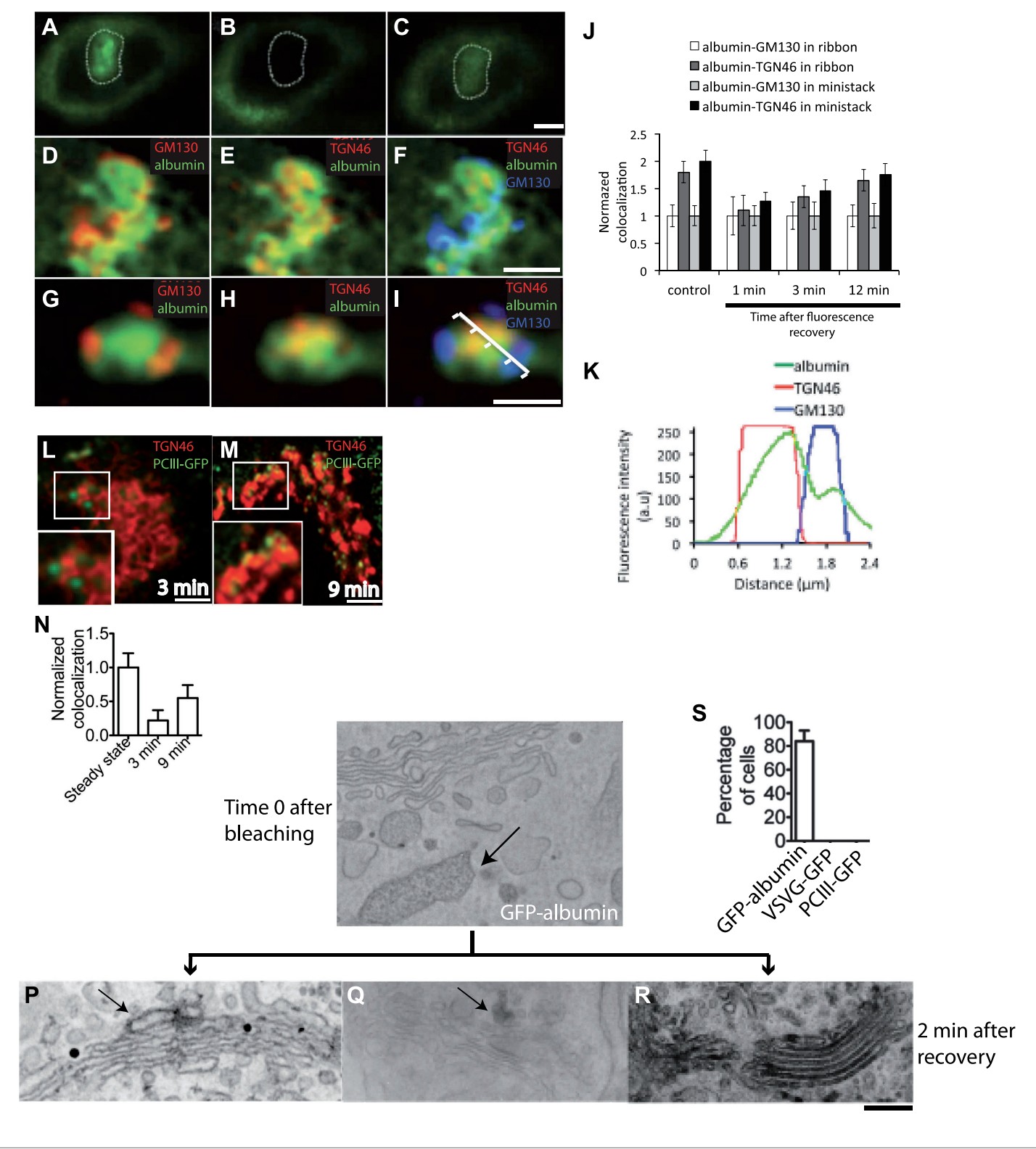

**Figure 3**. Kinetic patterns of transport of GFP-albumin, VSVG-GFP and PC-III-GFP through the Golgi stack under steady-state conditions. HeLa cells were transfected with GFP-albumin (**A–K**) or PC-III-GFP (**L–N**). After 16 hr of transfection, the Golgi area was bleached, and entry of these cargoes from the unbleached periphery (ER) into the Golgi area was monitored by FRAP. The cells were then fixed at different time points, stained for GM130 and

*Figure 3. Continued on next page*

*Figure 3. Continued*

TGN46, and re-localized for analysis of co-localization of the GFP-tagged cargoes with these Golgi markers. (**A–C**) Bleaching of the Golgi area, as delineated by the dotted line, with post-bleaching recovery for 1 min (**C**). (**D–F**) Detail of the same Golgi area shown in (**C**), showing co-localization of GFP-albumin (green) with GM130 (**D**, red), or TGN46 (**E**, red) or both (**F**: GM130, blue; TGN46, red). (**G–I**) Similar experiments carried out on a nocoda-zole-induced Golgi ministack ('Materials and methods'), with 1-min post-bleaching co-localization of GFP-albumin (green) with GM130 (**G**, red) or TGN46 (**H**, red) or both (GM130, blue and TGN46, red) (**I**). (**J**) Quantification of the degree of co-localization of GFP-albumin with GM130 and TGN46 at different time points after bleaching, as illustrated in (**A–F**). These data are expressed by normalizing the degree of co-localization of GFP-albumin in the TGN46 area to that of albumin in the GM130 area (set to 1). (**K**) Line scan along the arrow across the Golgi ministack shown in (**I**). The fluorescence intensities from representative points along the distance were plotted. (**L** and **M**) Cells were transfected with PC-III-GFP. The Golgi area (within the dotted line) was bleached, and the time course of entry of PC-III-GFP to the TGN was monitored. The cells were fixed and stained for TGN46 at 3 min (**L**) and 9 min (**M**) post-bleach, and the overlap between PC-III-GFP with TGN46 was examined. (**N**) Quantification of data in (**L** and **M**), expressed as mean ± SD from at least three independent experiments. (**O–S**) To ascertain the earlier observations of rapid filling of the Golgi stack by GFP-albumin (**A–F**), we resorted to electron microscopy. HeLa cells were transfected with GFP-albumin (**O** and **R**) or VSVG-GFP (**P**) or PC-III-GFP (**Q**). The Golgi localized fluorescence was bleached as before (time 0; **O**) and entry of cargo into the Golgi area monitored by FRAP and the cells fixed 2 min after recovery. The GFP fluorescence was then converted to a signal visible at the EM by photooxidation (see 'Photooxidation' under 'Materials and methods' section) using Diaminobenzidine (DAB). The DAB product is indicated by arrows. At time 0 the DAB product is present only in the ER with Golgi devoid of staining (**O**). After 2 min of fluorescence recovery, both VSVG-GFP (**P**) and PC-III-GFP (**Q**) are restricted to the cis-side of the Golgi, while GFP-albumin (**R**) is present throughout the Golgi. In the case of VSVG-GFP, DAB precipitate is visible outside of the Golgi cisternae because GFP is attached to the cytosolic tail of VSVG. In addition, nanogold labeling for Mannosidase II was done in (**P**) that marks the medial-part of the Golgi. The time 0 image shown is from cells expressing GFP-albumin; similar staining was obtained from both VSVG-GFP and PC-III-GFP expressing cells at time 0. (**S**) The percentage of cells that showed DAB product throughout the Golgi 2 min after recovery was calculated and presented as mean ± SD. Bar: 2 μm (**A–M**), 220 nm (**O–R**).

The following figure supplements are available for figure 3:

**Figure supplement 1**. Localization, transport behavior, and dynamics of GFP-albumin at steady-state.

**Figure supplement 2**. Kinetics of antitrypsin processing by Golgi enzymes reflects its fast kinetics of transport.

microscopy-based assays (*Figure 3—figure supplement 2*). In the same experiments, VSVG acquired endo-H resistance at a markedly slower rate (*Figure 3—figure supplement 2*). Thus, the Golgi residence time of soluble cargo appears to be sufficient for complete glycosylation. Possibly, the high surface/volume ratio of the flat Golgi cisternae maximizes the contact, and hence the efficiency of the reaction, between cargo and enzymes.

In summary, extensive kinetic evidence obtained under both traffic-synchronization and steady-state conditions shows the coexistence of two different intra-Golgi trafficking behaviors (and hence, presumably, different mechanisms) for different cargo types. PC-I (also PC-III) and VSVG enter the Golgi stack and move gradually from cis to trans, consistent with compartment progression, while albumin equilibrates rapidly across the Golgi compartments. The latter behavior is consistent with diffusion via intercisternal continuities. However, purely kinetic data cannot exclude that other mechanisms, such as fast vesicular shuttling, might lead to the same traffic pattern (*Pelham and Rothman, 2000*). To distinguish between diffusion- and vesicle-based traffic, we used both morphological and computational approaches.

## Albumin is depleted in Golgi vesicles and is present in Golgi intercisternal tubules

We first examined the Golgi structure in HepG2 cells at steady-state, with a focus on vesicles and tubules, using EM tomography. The Golgi stacks in HepG2 cells comprise 4–6 cisternae that were flanked by vesicles and connected side-by-side by 'longitudinal' tubules and fenestrated membranes, as seen in many other cell types. Some longitudinal tubules were Y-shaped and connected heterologous cisternae in neighboring stacks (not shown), in agreement with previous descriptions (*Marsh et al., 2004*). In addition, successive cisternae within individual stacks were sometimes connected by tubules that appeared to be oriented in the cis-trans direction ('vertical' tubules) and that were distributed apparently randomly at all levels of the stack, as previously described (*Trucco et al., 2004*). These vertical tubules were often convoluted (as exemplified in *Figure 4A–C* and *Video 1*), and had calibers ranging from 30 to 60 nm. The number of intercisternal connections between heterologous cisternae across the Golgi stack was highly variable. In HepG2 cells, each stack had about 5 ± 2 intercisternal connections (calculated as described in 'Materials and methods'). When a complete

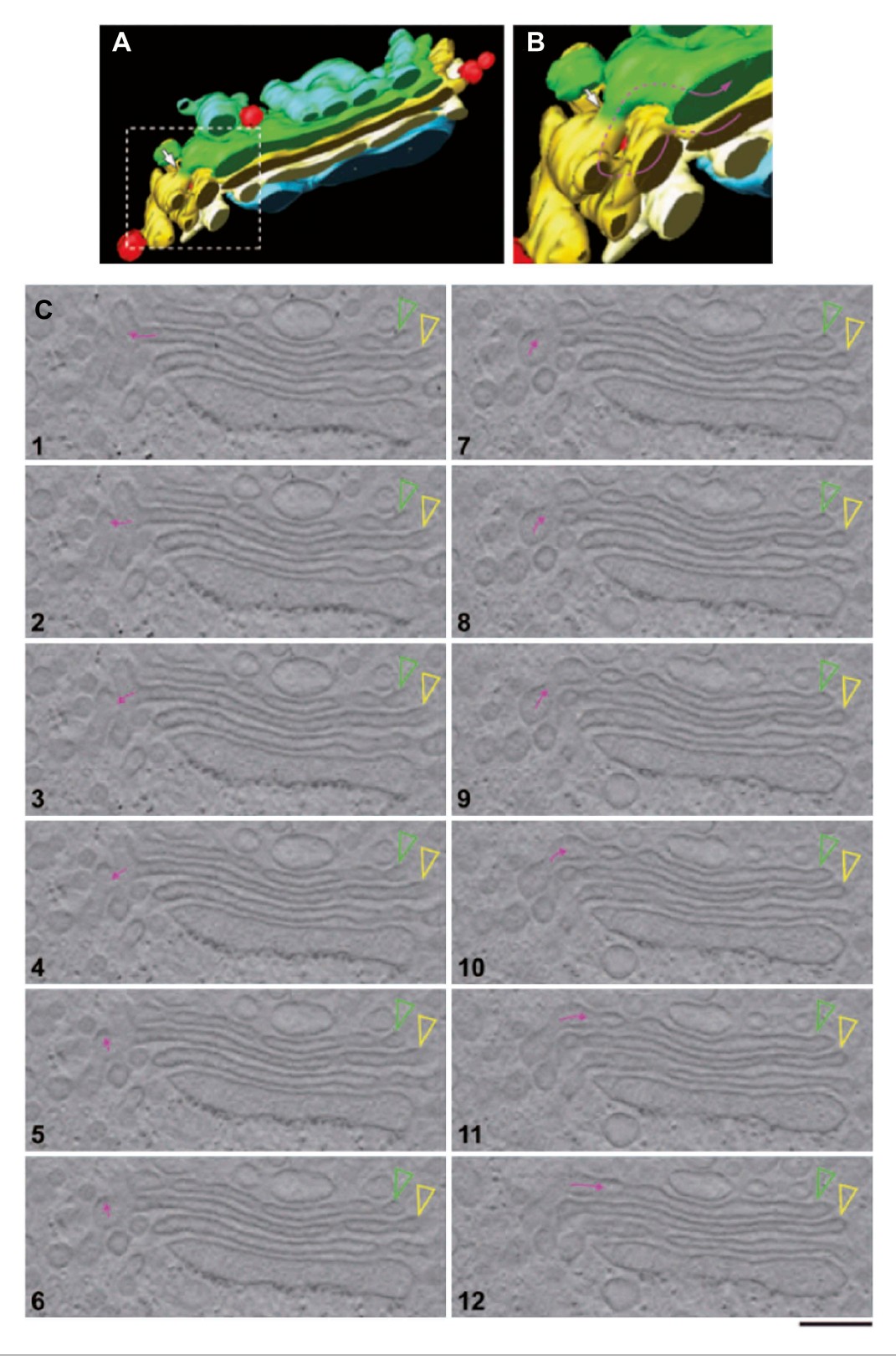

**Figure 4**. EM tomography facilitates the visualization of convoluted intercisternal tubules. HepG2 cells were high-pressure frozen and prepared for EM tomography ('Materials and methods'). (**A** and **B**) Tomographic model of a stack from a 200-nm-thick section containing an intercisternal connection. Detail of **A** shown in **B**; note the

*Figure 4. Continued on next page*

*Figure 4. Continued*

complexity of the convoluted connection (follow the arrow to identify the continuity). (**C**) A gallery of tomographic digital slices (panels 1–12) used to construct the model in (**A** and **B**) shows a convoluted intercisternal connection, with the small arrow following the connection, and the arrowheads showing the two cisternae that are connected. Note the complexity of the connection, which would be nearly impossible to detect in traditional thin sections. See *Video 1* for facilitated visualization of the continuity. Bar: 150 nm.

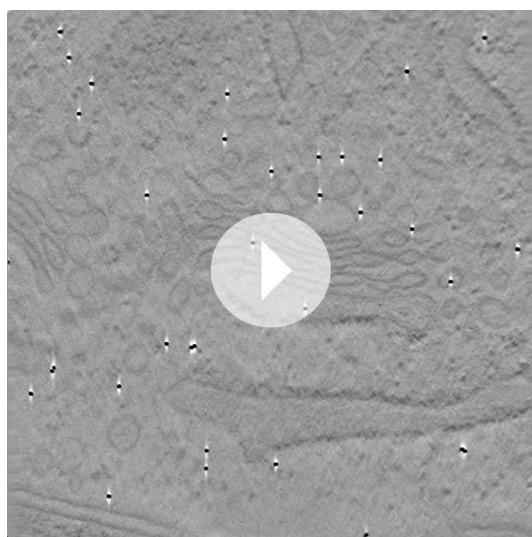

**Video 1**. A tomographic reconstruction of the Golgi stack shown in *Figure 4*. Scripts used to simulate the transport of albumin across the Golgi stack (Supplement to *Figure 6*):

tomographic reconstruction of a tubule was possible, it was always found that the tubule was connected to a cisterna (not shown) in agreement with previous observations (*Trucco et al., 2004*). A few narrow continuities joining the central areas of adjacent cisternae were also observed (not shown). Similar results were obtained using both chemical fixing and high-pressure freezing (not shown). For comparison, we also examined the Golgi in rat liver. The vertical and Y-shaped tubules were similar to those in HepG2 (not shown).

We then examined the albumin distribution in these Golgi structures by immuno-EM. Albumin was scarce in the ER (not shown) and dense in the stacks, and even denser in the TGN (*Figure 5A,C,G,I*). A similar distribution was also found in rat hepatocytes with a slightly higher concentration of albumin in Golgi and TGN (not shown). The distribution of antitrypsin was indistinguishable from that of albumin (not shown). For comparison, VSVG was markedly more concentrated in the TGN than in the stack (except in cells expressing high levels of VSVG, where the stacks were

filled with this cargo) (*Figure 5B,D,H,J*); and PC-I aggregates (in human fibroblasts) were also much more numerous in the TGN than in the Golgi stack (not shown). Thus, a large fraction of VSVG and PC-I in the Golgi area was located in the TGN. Next, we focused on the distribution of cargo in Golgi tubules and vesicles. We first examined the albumin content of Golgi vesicles (defined as round, 50–60 nm-wide profiles near the Golgi stack; see 'Materials and methods') by immuno-EM. The vesicular profiles were markedly depleted of albumin (*Figure 5E*, arrows, and 5k for quantification). This observation is in agreement with previous in vivo morphological and biochemical observations in animal liver cells (*Dahan et al., 1994*; *Gilchrist et al., 2006*) but is seemingly at variance with in vitro studies that showed that albumin can be present in vesicles generated by the non-hydrolysable GTP analogue GTPγS from Golgi enriched liver membranes (*Malhotra et al., 1989*) (see also [*Caro and Palade, 1964*; *Jamieson and Palade, 1967*]). As this difference might be due to the presence of GTPγS (instead of the natural nucleotide GTP) in the in vitro experiments (GTPγS is known to affect cargo sorting into vesicles [*Lanoix et al., 2001*]), we performed a series of in vitro experiments using GTPγS or GTP, and confirmed that this was indeed the case (*Figure 5—figure supplement 1*), explaining the discrepancy between the in vitro and the in vivo data.

We next probed the Golgi tubules. In random, thin sections for immuno-EM, tubules appear as elongated variably oriented tubular–ovoidal profiles in the vicinity of the stack ('Materials and methods'). These profiles contained albumin at a similar density to that seen in the cisternae (*Figure 5F*, arrowheads, quantified in K), independently of their orientation (parallel or perpendicular to the plane of the cisternae) (see below). For comparison, we also examined VSVG. This cargo has been previously shown to be depleted in Golgi vesicles and, to a lesser extent, in tubules (*Trucco et al., 2004*). We confirmed that VSVG is lower in both vesicular and tubular peri-Golgi profiles than in

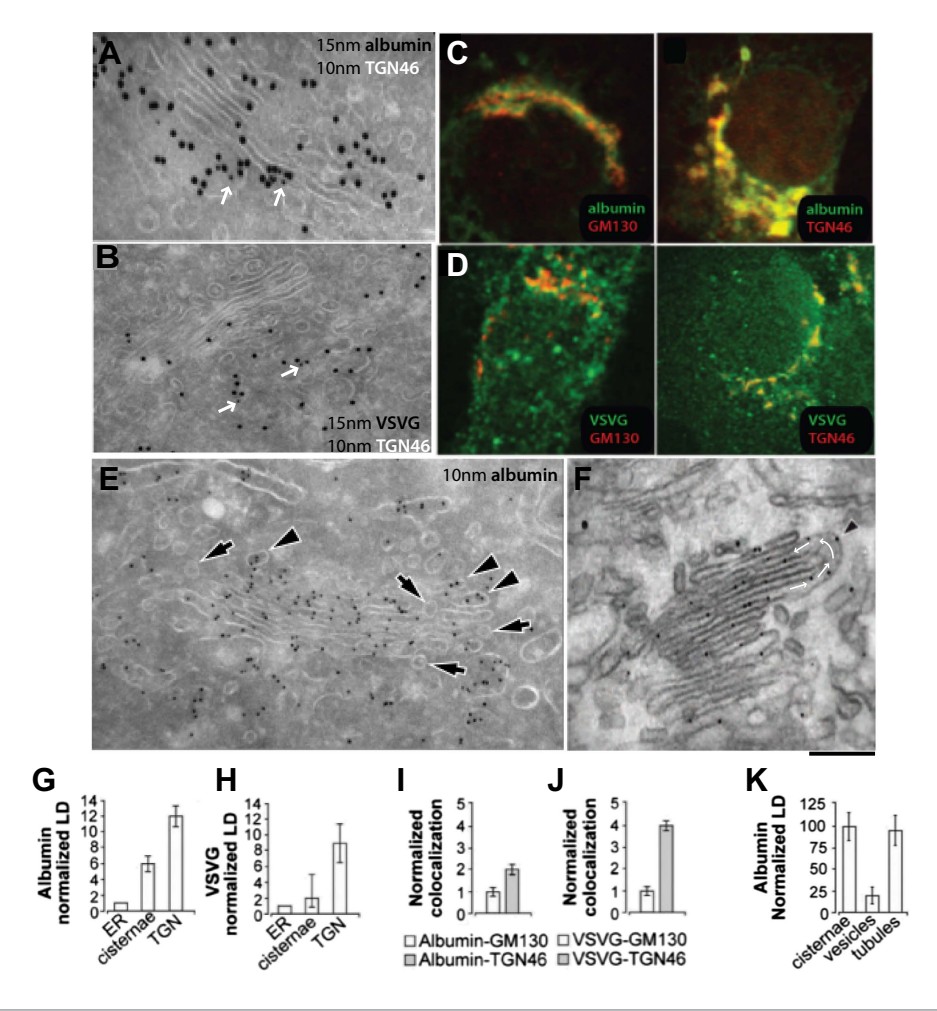

**Figure 5**. Albumin distribution in Golgi cisternae, vesicles and tubules in HepG2 cells. (**A–D**) Albumin and VSVG distribution in the ER, Golgi stack and TGN (**A** and **B**, immuno-EM; **C** and **D**, immunofluorescence). White arrows in (**A** and **B**) indicate TGN46 labeling. For quantification see (**G–J**). (**E**) Albumin distribution in cisternae, vesicles (50–60 nm wide round profiles near the Golgi stack; arrows) and tubules (tubular–ovoid profiles; arrowheads), by immuno-EM. (**F**) Albumin present in a connected cis-trans vertical tubule, as visualized by the immuno-nanogold technique. Arrowhead indicates albumin and white arrows highlight the intercisternal connection. For quantification see (**K**). (**G** and **H**) Quantification of labeling density (LD) of albumin and VSVG by immuno-EM, normalized to ER labeling. (**I** and **J**) Quantification of co-localization (as described in 'Materials and methods') of the cargoes with GM130 (cis-Golgi) and TGN46 (TGN) markers by immunofluorescence, normalized to their co-localization with GM130. (**K**) Quantification of distribution of albumin in cisternae, peri-Golgi vesicles and tubules expressed as LD. Bars: 120 nm (**A**), 210 nm (**B**), 7.5 µm (**C** and **D**), 250 nm (**E** and **F**).

The following figure supplements are available for figure 5:

**Figure supplement 1**. Distribution of cargoes in COPI vesicles.

**Figure supplement 2**. Gallery of cryo-immuno-gold EM images indicating the presence of albumin in intercisternal tubules.

**Figure supplement 3**. Presence of albumin in intercisternal tubules revealed by serial sectioning followed by cryo-immuno-gold EM and DAB photooxidation followed by tomography.

**Figure supplement 4**. Presence of albumin in intercisternal tubules revealed by cryo-immuno EM followed by tomography.

cisternae (*Figure 5—figure supplement 1*). We then attempted to visualize albumin in complete vertical tubular connections. This experiment presented serious difficulties because: (a) most connections are convoluted and cannot be included in single thin sections; and, (b) even relatively 'linear' continuities are very unlikely to be cut through their entire length at random (here, a reasonable expectation is that less than one connection may be found in hundred sections, assuming six connections per stack [*Marsh et al., 2004*; *Trucco et al., 2004*]). An additional difficulty is that the tubular membranes are often cut obliquely at some point along their length, resulting in defective membrane visualization. We sought to overcome these problems by (a) cutting several hundreds of individual thin sections, as well as several serial sections, to find at least a few complete inter-cisternal connections, and, (b) combining tomography with albumin labeling by photo-oxidation and cryo-immuno EM. Using the first approach, we succeeded in visualizing a few tubular connections (*Figure 5—figure supplements 2 and 3*) that showed continuity across heterologous cisternae (see legend to these figures). These connections contained albumin at roughly the same level as in cisternae (*Figure 5F*, *Figure 5—figure supplements 2 and 3*). Moreover, the tomography- and photooxidation-based approaches (*Meiblitzer-Ruppitsch et al., 2008*) also indicated that albumin is present in the connecting tubules (*Figure 5—figure supplements 3 and 4*).

In sum, despite the technical difficulties, the data are consistent with the notion that albumin is depleted in peri-Golgi vesicles and it is present in Golgi intercisternal tubules at levels similar to those seen in cisternae.

## Computational comparison of vesicle-based with diffusion-based models of intra-Golgi transport

To further distinguish between continuity-based and vesicle-based albumin transport, we then used two computational models. These models were constructed to assess whether the equilibration of albumin at the observed rates (within 1–2 min) through a closed system with a stack-like geometry (defined using morphological parameters derived from our observations) could be best explained by a scheme based on simple diffusion of albumin between the cisternae through tubules without biasing forces, or by a scheme based on vesicular transport. Notably, these models are limited to simulating intra-Golgi equilibration of cargoes and do not aim to simulate the entire traffic process through the Golgi including cargo arrival, departure and intra Golgi concentration steps.

For diffusion-based transport, we simulated albumin diffusion through either one stack or through Golgi ribbons made of three to five longitudinally connected stacks containing vertical intercisternal tubules ('Materials and methods' and *Figure 6A,B*). Different numbers, dispositions and stabilities (open time) of these tubules were tested in the simulation (*Table 1*; *Figure 6C–F*). We found that the rates of equilibration from cis to trans Golgi are fast, and are easily compatible with our experimental data (see *Figures 1–3* and *Figure 1—figure supplement 1* for experimental data), even when we simulated infrequent and transient tubular connections (*Figure 6C*). Also of note, even when connectivity gaps at any level of a stack are simulated, these gaps can be compensated for by the presence of connections at the same level in neighboring stacks longitudinally joined in Golgi ribbons, and/or by the fact that these gaps might be transient, i.e., that connections might form and disassemble rapidly (see scheme in *Figure 6D–F*).

For vesicle-based models (*Figure 6—figure supplement 1*; *Table 2*), we implemented a quasi-hopping scheme where one 'event' includes the loading and unbinding of a vesicle from one cisterna and binding and unloading of the same vesicle to a neighboring cisterna. This scenario is the limiting case allowing for fastest turnover. The considered system had the same geometry and number of cisternae as those used for the diffusion-based model (*Figure 6*). We found that in order to mediate cargo equilibration across the Golgi stack in 2 min, each cisterna would need to generate and receive hundreds of vesicles per second (the exact number varies depending on the variables selected for simulation, e.g., size of the vesicles or the number of cisternae per stack etc, see *Figure 6—figure supplement 1*; *Table 2*). This very high rate of turnover is very difficult to reconcile with several lines of experimental evidence, as discussed in detail in the legend to *Figure 6—figure supplement 1* and in the 'Discussion'.

In conclusion, the observed rate of intra-Golgi albumin transport (cis-trans equilibration is observed in less than 2 min), seems easily compatible with diffusion-based models and inconsistent with vesicle-based quantitative transport models of albumin equilibration through the stack (see legend to *Figure 6—figure supplement 1* and 'Discussion').

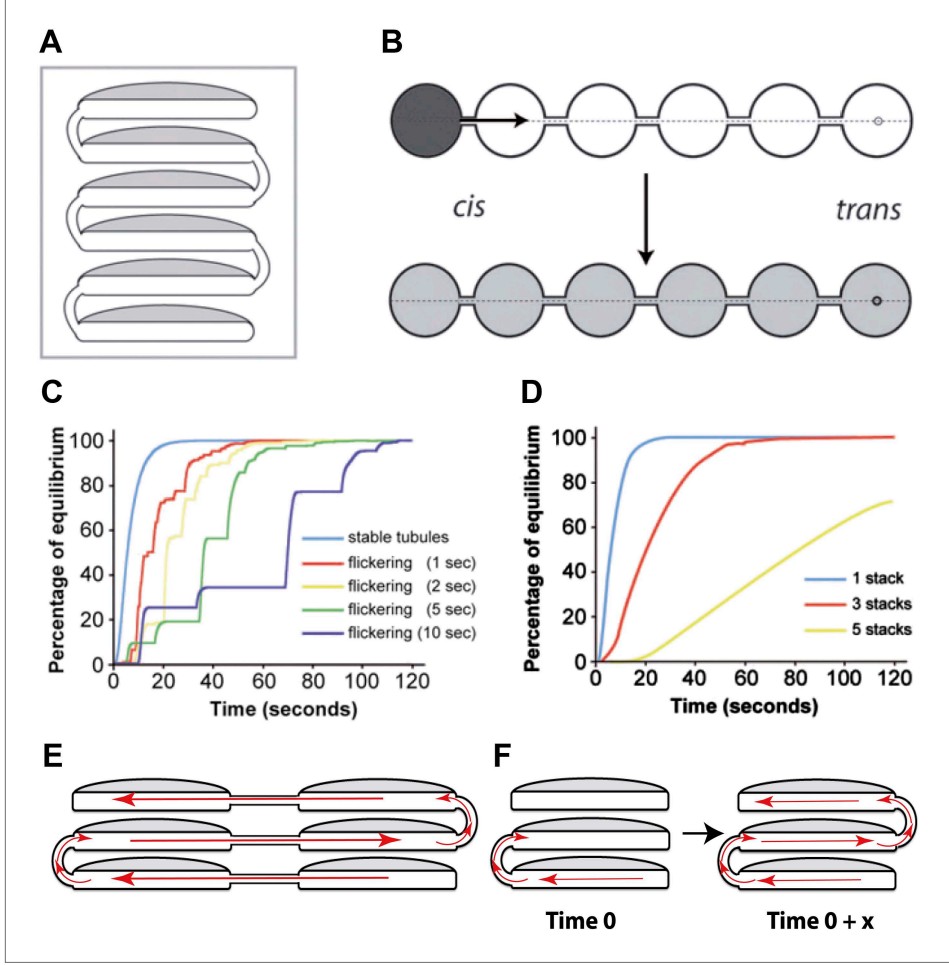

**Figure 6**. Computational simulations of intra-Golgi transport of albumin by diffusion via intercisternal tubules. (**A**) The Golgi stack was modeled as a system of six circular cisternae connected in series by five (one per pair of cisternae) vertical cylindrical tubules. (**B**) The same stack drawn in a 'distended' disposition. The size of the cisternae was set to 1.5 μm diameter and 30 nm thickness, and the diameter and length of the tubules to 30 nm and 100 nm, respectively. The simulations started with the first cisterna (*cis*) filled with albumin (black shading) and all of the others empty (no shading). Albumin was then allowed to diffuse through the connections until it asymptotically reached equilibrium (gray shading in all cisternae). The variations with time in the albumin concentrations in the other cisternae were calculated for the center of the cisternae (see below). For wider tubule diameters of 60 nm and 120 nm, the 90% threshold of equilibrium was reached after 7.8 s and 6.8 s compared to 14.9 s for the 30 nm diameter. For shorter and longer tubules of 30 nm diameter with 50 nm and 150 nm lengths, the 90% threshold of equilibrium was reached after 11.4 s and 19.1 s. When assuming a system composed only of four circular cisternae, the 90% threshold of equilibrium in the fourth cisternae was reached after 3.4 s (60 nm diameter, 100 nm length) and 3.0 s (120 nm diameter, 100 nm length) compared to 6.4 s for the 30 nm diameter tubules. For shorter and longer tubules of 30 nm diameter with 50 nm and 150 nm lengths, the 90% thresholds of equilibrium were reached after 5.0 s and 8.4 s. (**C** and **D**) Time-courses of the equilibration process with different Golgi configurations and with stable or transient (flickering) intercisternal tubular connections. (**C**) One stack of six cisternae connected by one stable tubule per pair of adjacent cisternae (five connections in all), or by one transient tubule per pair of adjacent cisternae. The tubules were set to be open for 50% of their time, with equal average open and closed times as indicated. In these simulations, the individual tubules opened and closed randomly. (**D**) Simulation of diffusion based albumin transport in Golgi ribbons of one (blue), three (red), or five (yellow) stacks (each with 6 cisternae). The ribbons were completely connected horizontally by tubules joining adjacent cisternae. The total number of connections was five in all cases. For instance, the three-stack ribbon had one or two connections per stack. Nevertheless, complete equilibration was reached in less than 60 s. (**E**) Possible diffusion route of a soluble cargo through the Golgi ribbon with three stacks where the longitudinal tubules connecting the isolated

*Figure 6. Continued on next page*

*Figure 6. Continued*

stacks compensate for the vertical connectivity gaps. (**F**) Diffusion of a soluble cargo across a stack through transient intercisternal tubules.

The following figure supplements are available for figure 6:

**Figure supplement 1**. Computational simulation of the intra-Golgi equilibration of albumin argues against the classic vesicular transport model.

**Figure supplement 2**. Computer-simulated model of the formation of a pH gradient between continuous *cis*- and *trans*-Golgi cisternae.

## Discussion

The main findings of this study are that soluble cargo proteins such as albumin traverse the Golgi complex rapidly by a mechanism that is different from compartment progression and involves diffusion via intercisternal continuities. This fast diffusion-based transport coexists in the same Golgi stacks with the slower movement of supramolecular cargo such as PC by compartment progression.

Three lines of evidence (kinetic, morphological, and computational) converge to support the notion of continuity-based transport. The kinetic evidence, which consists of the observation that albumin traverses the stack in less than 2 min, at least fivefold to sixfold faster than PC, is consistent with diffusional transport, a fast process over short distances. It cannot, however, exclude in principle a role for other (potentially) rapid transport processes such as intercisternal shuttling by COPI-derived vesicles (**Pelham and Rothman, 2000**; **Presley et al., 2002**). Thus, further discriminating evidence is required. The morphological experiments show that Golgi vesicles are depleted (80%) of albumin while Golgi intercisternal tubules appear to contain albumin at similar levels as those seen in cisternae. Published biochemical studies concur with this depletion in vesicles (**Dahan et al., 1994**; **Sonnichsen et al., 1996**; **Gilchrist et al., 2006**) indicating that the observed depletion is not an artifact of reduced epitope presentation in the vesicles. These observations therefore favor a role for tubular continuities, over vesicles, in albumin transport. Clearly, the albumin depletion in vesicles is difficult to reconcile with the extreme efficiency of the intra-Golgi albumin transport. Moreover, computational models of

**Table 1.** Simulated intra-Golgi transport of albumin by diffusion, via intercisternal continuities, occurs in the timescale of seconds

| Length of tubule (nm) | Diameter of tubule (nm) | Number of cisterna | Time needed for 90% equilibration (s) |
|---|---|---|---|
| 100 | 30 | 6 | 14.9 |
| 100 | 60 | 6 | 7.8 |
| 100 | 120 | 6 | 6.8 |
| 50 | 30 | 6 | 11.4 |
| 150 | 30 | 6 | 19.1 |
| 100 | 30 | 4 | 6.4 |
| 100 | 60 | 4 | 3.4 |
| 100 | 120 | 4 | 3.0 |
| 50 | 30 | 4 | 5.0 |
| 150 | 30 | 4 | 8.4 |

The size and geometry of the Golgi stack used for the simulations is defined in the legend to **Figure 6**. The variable parameters used for the simulation are: length of tubules (from 50 to 150 nm), diameter of tubules (from 30 to 120 nm), and number of cisternae (between 4 and 6). The tubules here refer to the intercisternal tubules connecting two cisternae of a Golgi stack. The time needed for 90% equilibration of albumin across the Golgi stack under varying combinations of the indicated parameters was computed. As can be seen from the data, equilibration across the cisternae happens in seconds across all the conditions.

**Table 2.** Simulated intra-Golgi transport of albumin mediated by vesicles, predicts extremely fast turnover of cisternal rims

| Vesicle diameter (nm) | 50 | | 60 | | 70 | |
|---|---|---|---|---|---|---|
| Number of cisternae | 4 | 6 | 4 | 6 | 4 | 6 |
| Steps for 90% equilibration | 19,617 | 44,230 | 11,352 | 25,595 | 7149 | 16,120 |
| Rim turnover time (s) | 0.576 | 0.255 | 0.829 | 0.368 | 1.129 | 0.500 |

The size and geometry of the Golgi stack and vesicles used for the simulations are defined in the legend to *Figure 6—figure supplement 1*. The variable parameters used for the simulation are: diameter of vesicles (from 50 to 70 nm) and number of cisternae (between 4 and 6). Albumin transport is considered to proceed stepwise, where each step is defined as one vesicle detaching from each cisterna and fusing with an adjacent one. For the calculations presented here, the albumin concentration in the vesicles is considered to be 20% of that present in the cisterna (see *Figure 5* and also legend to *Figure 6—figure supplement 1*). The number of steps required to achieve 90% equilibration of albumin across the stack was computed and the time required for a single turnover event of the cisternal rim (rim turnover time) was calculated as described in the legend to *Figure 6—figure supplement 1*. The rim turnover time varied from 0.25 to 1.12 s or in other words, the rim turns over from 4 to 1 times per second, depending on the condition used for the simulation. The results presented here are for the scenario 'a' (discussed in *Figure 6—figure supplement 1*) and the results are very similar even in the case of scenario 'b'.

intra-Golgi transport by diffusion show that tubules, even when very few and transient, are sufficient to mediate the observed fast rate of albumin transport through the stack. Instead, models of vesicular transport show that Golgi vesicles would have to form and fuse at rates of hundreds of vesicles per cisterna per second, and hence that cisternal rims would have to turnover several times per second, to mediate albumin transport (see legend to *Figure 6—figure supplement 1* for a detailed discussion of this point). These computational results appear difficult to reconcile with at least two lines of experimental evidence: (a) with such rates of vesicle formation/fusion, there should be morphological signs of such an enormous activity; however, cisternae look quiet, with fairly rare images of budding or fusion; and, (b) if cisternal rims turned over a few times per second as a result of vesicle budding, then the vesicular coat protein COPI (or at least a substantial subpopulation of COPI) should cycle on and off the Golgi complex at similar rates, i.e., roughly three times per second; however, the half-time of COPI dissociation from the Golgi complex is roughly 35 s in live cells (*Presley et al., 2002*). Finally, the vesicle turnover rates calculated to be required for albumin transport are orders of magnitude higher than those experimentally observed for any other type of coated vesicles at any cellular location including endocytic vesicles (about 2 pinocytic vesicles per second per macrophage cell [*Steinman et al., 1976*]) or fast neuronal synapses that form close to 1–15 endocytic vesicles per second per synapse (*Rink et al., 2005*; *Fernandez-Alfonso and Ryan, 2006*) or COPII vesicles (about 0.3 vesicles per exit site per second [*Thor et al., 2009*]).

Collectively, these considerations indicate that the simplest scheme to accommodate all of the available evidence on albumin intra-Golgi transport is one in which this soluble cargo reaches the cis-Golgi via the IC and diffuses rapidly across the Golgi stack in the cis-trans direction through (most probably transient) intercisternal tubules and concentrates on the trans-side. At the same time, non-diffusible cargoes that cannot enter intercisternal tubules (such as PC-I) traverse the stack slowly, by progression–maturation (*Mironov et al., 2001*). As noted earlier, continuity-mediated transport has been proposed to occur also in the endo-lysosomal pathway (*Luzio et al., 2007*) and during the release of synaptic vesicles through flickering pores (*Rizzoli and Jahn, 2007*) (*Alabi and Tsien, 2013*).

Among the questions raised by the cargo diffusion model, one concerns the functional significance of its coexistence with maturation. We suggest that each process optimizes the transport of different cargo classes endowed with specific physical properties. For instance, PC-I forms aggregates in the early Golgi (or earlier) and is efficiently transported in this condensed state by compartment progression. Albumin and possibly other abundant serum proteins, instead, move from the ER to the Golgi in a soluble state (*Martinez-Menarguez et al., 1999*; *Oprins et al., 2001*). Thus, once at the Golgi, these proteins diffuse rapidly via continuities away from the cis-Golgi towards the TGN, where they are concentrated and exported. Existence of different modes of transport possibly also gives the

cell/organism flexibility to switch between modes of transport depending on the cargoes being transported in a certain physiological or developmental stage. For example, during the spermatid development there is a clear increase in the proliferation of intracisternal tubules (*Vivero-Salmeron et al., 2008*) that we propose may coincide with an increase in the transport of cargoes that depend on diffusional mode of transport.

The second question raised by the cargo diffusion model is how the Golgi maintains its compositional polarity in the presence of continuities and does not collapse to form one larger compartment. Several possibilities can be hypothesized. For instance, the collapse of connected cisternae may be prevented by scaffolds that maintain cisternal geometry, or it might actually tend to occur, but at a rate that is too slow compared to the half-life of transient continuities. Regarding domain segregation within continuous compartments, there are several known prior examples. They include the ER, with its rough and smooth domains (*Sitia and Meldolesi, 1992*), the neuronal plasma membrane, with different compositions in the soma and dendrites (*Bradke and Dotti, 2000*), and even the cytosol, where gradients of calcium and other second messengers are continuously created and maintained across different cytosolic regions (*Zaccolo et al., 2002*). In most cases, the molecular basis for domain segregation is ill-understood, with a few possible exceptions (*Zaccolo et al., 2002*). For the Golgi complex, multiple such domain-generating mechanisms might exist, and they might be based on: (a) the small size and the transiency of intercisternal tubules, which might act as filters by remaining open for a time sufficient for the passage of molecules such as albumin that would rapidly diffuse through the tubes, but not for that of larger molecules, such as lipid rafts, or protein clusters, which would be retarded by the small size and curvature of the tubes; (b) the arrival at the cis- and trans-Golgi poles of membranes of markedly different compositions and thicknesses from the ER and from endosomes, with the consequent creation of a cis-to-trans lipid compositional gradient through which Golgi resident enzymes may distribute differentially (*Trucco et al., 2004*; *Patterson et al., 2008*); (c) the action of cytosolic scaffolds to nucleate different domains in different cisternae; and, (d) the action of intralumenal buffering and pumping systems to create gradients of ions, such as calcium and protons, in a fashion similar to that described for cytosolic messengers (*McCarron et al., 2006*) (a quantitative model of this last mechanism and a further discussion of diffusion via continuities are in *Figure 6—figure supplement 2*). Clearly, much work is needed to clarify whether and how some of these mechanisms apply.

Another point of consideration while discussing the cargo diffusion model is the possible relationship between intra-Golgi transport and Golgi export, although a detailed analysis of the mode of albumin export is beyond the scope of this study. Earlier biochemical and our own imaging studies have shown that small soluble cargoes are cleared from the cells quite rapidly and a pH gradient is necessary for such an efficient export (*Yilla et al., 1993*) (our unpublished data). Moreover, we show that albumin is concentrated in the TGN (*Figure 5G*), and that this concentration most likely depends on the low TGN pH, as it is abolished by concanamycin, a specific inhibitor of the vacuolar proton pump that operates in the TGN (not shown). Thus, the pH-dependent concentration of albumin on the trans-side of the stack (*Figure 5G*) appears to be an important driving force for albumin export in that it imparts some sort of directionality to the movement of this cargo by increasing the probability of cargo molecules to localize in the TGN vs the CGN. Based on the above collective considerations and the current results, a hypothetical transport model for albumin might be delineated as follows: albumin reaches rapidly the Golgi from the ER, where it diffuses rapidly across the stack at a rate dependent on the number and transiency of the connections, accumulating in the low pH trans-Golgi compartments. This intra-Golgi equilibration of albumin is fast and hence unlikely to be rate-limiting for export out of the Golgi (differing in this regard from the slower transport of larger cargo such as PC-I which takes 10–15 min to cross and exit the stack). Rather, export is likely to depend on the rate of formation of the export carriers and on the concentration of albumin in these carriers, which, as noted, depends on the TGN pH. A notable consequence of this is that the export efficiency of other soluble cargoes might vary as a function of their propensity to concentrate in the TGN at low pH. The mechanism by which a low pH increases the albumin concentration in the TGN is unknown. On a speculative plane, at low pH, albumin might bind with increased affinity to a TGN protein or lipid, resulting in enhanced concentration and sorting into export carriers. Under these conditions albumin might also tend to self-aggregate, increasing concentration, and sorting efficiency. Interestingly, under the vesicular transport model, concentration in the TGN could be achieved by directional vesicle-mediated transport in a pH-independent way. Thus, the fact that export is pH-dependent is a further indication in favor of the diffusion-based mechanism.

Finally, one may ask whether the coexistence of diffusion- and maturation-based mechanisms might help to rationalize the observation of different intra-Golgi trafficking patterns for various cargo proteins and under different conditions. Clearly, the trafficking of large non-diffusible cargoes (e.g., PC-I) and that of soluble proteins (e.g., albumin) is explained in a simple way by the above dual transport scheme. The case of VSVG, a trimeric transmembrane protein, is more complex, in that this cargo has been reported to either progress gradually through the stack like PC-I (here and *Mironov et al., 2001*; *Trucco et al., 2004*), or to spread rapidly from cis to trans cisternae like albumin (*Bergmann and Singer, 1983*; *Patterson et al., 2008*), depending on experimental conditions. Here, a simple explanation would be that this dual behavior might depend on the ability of VSVG to exist in two states, an 'aggregated' one, in which it would form large clusters that behave like PC-I (as proposed by others *Griffiths et al., 1985*), and a diffusible one (mono/oligomers) that behaves like albumin. Under this assumption, which remains to be verified, also the dual behavior of VSVG could be accommodated by our two-mechanisms scheme.

The coexistence of multiple transport principles, involves a loss of simplicity and elegance compared to a single general mechanism. Nevertheless, evidence that multiple cargo transfer strategies are used in the secretory and endo-lysosomal pathways is emerging (*Luzio et al., 2007*) and the observation of different cargo transport rates for different cargoes (*Boncompain et al., 2012*) concurs well with the notion of multiple transport mechanisms. It is unclear whether the mechanisms so far described represent the full range of the existing transport strategies. The complete scenario will probably emerge through studies of further cargoes and trafficking steps in suitable cell types. Highly secreting cells, including certain cancer lines, might preferentially use one rather than another traffic mechanism. Uncovering the diversity of the trafficking modes will enhance our ability to understand and selectively manipulate different cargo classes, for research or therapy purposes.

## Materials and methods

### Cells, DNA constructs, antibodies, and reagents

HepG2 human hepatoma and HeLa cells (both from ATCC) were grown in Minimum Essential Medium (MEM) supplemented with 10% foetal calf serum, glutamine and antibiotics. Human fibroblasts were grown and used as described previously (*Mironov et al., 2001*). The following polyclonal antibodies were used: anti-GM130 (MA De Matteis, TIGEM, Italy), anti-TGN46 (S Ponnambalam, Leeds University, UK), anti-albumin and anti-α1-antitrypsin (DAKO, Denmark), and anti-VSVG (MA De Matteis). The following were also used: nanogold-conjugated Fab fragments of anti-rabbit IgG and Gold Enhancer (Nanoprobes, Yaphank, NY); Protein A conjugated with colloidal gold (J Slot, Utrecht University, The Netherlands); and anti-rabbit, anti-mouse and anti-sheep antibodies conjugated with Alexa Fluor 488, Alexa Fluor 546 and Alexa Fluor 633 (Molecular Probes Europe BV, The Netherlands). GTPγS and CHX were from Sigma (St. Louis, MO). The FUGENE6 transfection reagent (used following the manufacturer instructions) was from Roche (Basel, Switzerland). To express constructs in human fibroblasts (which were difficult to transfect), microinjection was performed. Albumin was expressed efficiently in 50% of injected cells. Of these, 30% expressed both albumin and PC, while the other 70% expressed only albumin. The other 50% of the injected cells contained PC and very little or no albumin, indicating that these human fibroblasts prefer to express either one or the other of these cargoes. Unless otherwise indicated, all other chemicals and reagents were obtained from previously described sources (*Mironov et al., 2001*).

The GFP-albumin construct was prepared by PCR amplification and sequential cloning of the pre-pro-signal region of albumin, the GFP cDNA from the pEGFP-N2 vector (Clontech laboratories, Takara Bio Europe), and the albumin without the signal region, into the pcDNA-4B vector. Specifically, the construct was prepared by subcloning the HindIII-BamHI-digested, PCR-amplified (forward primer, CCCAAGCTTATGAAGTGGGTAACCTTTATTTCCC; reverse primer CGCGGATCCTCGACGAAACACA CCCCTGG) pre-pro-albumin region into the pcDNA-4B vector, followed by the sub-cloning of the BamHI-EcoRI-digested, PCR-amplified (forward primer, CGCGGATCCGTGAGCAAGGGCGAGGAGC; reverse primer, CCGGAATTCCTTATACAGCTCGTCCATGCCGAG) GFP cDNA into the pre-pro-albumin-containing construct, and in turn the sub-cloning of the EcoRI-XhoI-digested, PCR-amplified albumin (forward primer, CCGGAATTCGATGCACACAAGAGTGACCTTC; reverse primer, CCGCTCGAGTTA TAAGCCTAAGGCAGCTTGAC) into the previous construct. The final construct was verified by direct sequencing. The preparation the PC-III construct was as described previously (*Perinetti et al., 2009*).

## Cell treatments and transport assays

### Traffic synchronization protocols

#### Synchronization protocol 1

(32–15°C): Human fibroblasts were microinjected with a plasmid of interest and then shifted to 15°C for 2 hr to accumulate cargo in the IC, and then shifted back to 32°C to monitor cargo transit through the secretory pathway. The cells were then fixed at increasing time interval and processed for IF or EM.

#### Synchronization protocol 2

(CHX/32-15°C): HepG2 cells were infected with VSV (1 hr at 32°C) and then kept at 32°C for 3 hr in the presence of CHX. CHX was washed out and the cells were placed at 15°C for 2–3 hr to allow the synthesis of new proteins and their transport to the IC. Finally, the cells were shifted to 37°C or 32°C in the presence of CHX, to allow the passage of cargo proteins through the Golgi complex. The cells were then fixed at increasing time interval and processed for IF or EM. Of note, CHX does not affect the trafficking rates, as established using the pulse-chase transport assay.

### FRAP-based ER-to-Golgi transport assay

HeLa cells (25,000–30,000) were seeded in MatTek CELLocate dishes, transfected with GFP-albumin, and kept at 37°C for 16 hr. To determine the rate of GFP-albumin entry into the Golgi complex from the ER at steady-state, the Golgi area was photobleached and the rate of fluorescence recovery into this area was monitored. It is known that after photobleaching, GFP can recover its original conformation and fluorescence, albeit partially and slowly. To verify whether this phenomenon might contribute to the Golgi FRAP and thus 'contaminate' the signal due to GFP-albumin entry into the Golgi complex, we bleached whole cells and monitored the recovery of fluorescence. This recovery was low and contributed to less than 5% of the Golgi FRAP during the time-scale of the experiments. For each cell, the intensity of the Golgi area was recorded and normalized to the intensity of the whole cell, to account for the overall intensity loss during bleaching. The same approach was used to measure the diffusion of albumin along the Golgi ribbon. Since the rate of diffusion of albumin along the Golgi ribbon was an order of magnitude faster than the ER to Golgi transport of albumin (*Figure 3—figure supplement 1*), we have assumed that the contribution of the latter is negligible. For exit from the Golgi complex, the FRAP-related iFRAP technique was used (*Patterson et al., 2008*).

### Intra-Golgi transport assay by correlative video-light microscopy

The cells were handled as above. The Golgi complex was photobleached (with 100% transmission of laser and 50 iterations) and the Golgi FRAP was monitored. At suitable intervals after bleaching, the cells were fixed with 0.05% gluteraldehyde in 150 mM HEPES for 5 min, and then processed for immunofluorescence (i.e., labeled for the Golgi markers GM130 and TGN46). It is important to note here that fixation of monolayers of cells is a rapid process that happens in a time frame 1–2 s (*Polishchuk et al., 2000*), and so any artifact due to slow fixation of the samples is nonexistent. This procedure prevented the recovery of fluorescence of the GFP by spontaneous recovery of its fluorescent conformation (see above). The same cells that had been photobleached and monitored were then identified by their spatial coordinates (*Mironov and Beznoussenko, 2013*) and the degree of co-localization between GFP-albumin that had re-entered the Golgi and the Golgi marker proteins GM130 and TGN46 was determined (see below).

## Confocal microscopy and co-localization analysis by immunofluorescence

Confocal and time-lapse images were obtained using a Zeiss LSM510 META confocal system (Carl Zeiss, Jena, Germany). To measure co-localization between cargo and the Golgi marker proteins GM130 and TGN46, we used approaches modified from published protocols (*Mironov et al., 2001*). Co-localization indicates the intensity of cargo staining within the GM130- or TGN46-stained areas (cargo localized in other cellular or Golgi regions was not considered). Two methods were used:

### Method 1

This method is a modification of the weighted co-localization analysis of the Zeiss LSM510 META confocal image analysis software, according to *Manders et al. (1993)*. It involves subtracting the background in each channel. The standard Zeiss value for background is arbitrarily set at 40% of the peak

value, but this can be changed to reduce overlap between channels. This method was used in parallel with immuno-EM and stereology experiments, and gave results that were in good agreement with the immuno-EM data. Note that because of the resolution limit of the optical system, with this method the degree of overlap calculated between the Golgi markers themselves (GM130 and TGN46, which are in reality completely segregated in the cis-Golgi and the TGN, respectively) is about 20% in HepG2 cells under optimal conditions. This means that the calculated distribution of GM130 itself between the cis-Golgi and the TGN is about 80% in the cis-Golgi and 20% in the TGN. Vice versa, TGN46 is calculated to be about 20% in the cis-Golgi and 80% in the TGN. This defines the maximal dynamic range of the method: a cargo moving from the cis-Golgi to the TGN will shift from a GM130/TGN46 ratio of about 80/20 (in the cis-Golgi) to a ratio of 20/80 (in the TGN). This applies to HepG2 and HeLa cells; however, under specific conditions (i.e., when the Golgi complex is fragmented into separated stacks, and especially in some stacks, which are presumably those oriented with cisternal planes parallel to the beam), the resolution between GM130 and TGN46 can be greater.

## Method 2

To eliminate the arbitrariness of the background subtraction in method 1, we developed a modification of the co-localization approach described by *Manders et al. (1993)*, using the MATLAB software (The MathWorks, Inc., R2006b). Briefly, an optical section taken through the center of the Golgi region was recorded and the intensities of the cis-Golgi and trans-Golgi markers and of the cargo were normalized to the maximum of the corresponding channel for each pixel. To calculate three-color co-localizations, i.e., the co-localization of cargo with each of the *cis*-Golgi and trans-Golgi markers, the intensity distributions were interpreted as probability distributions for the specific markers. Thus, the probability for a cargo protein to co-localize with a cis-Golgi or *trans*-Golgi marker was calculated for each pixel by multiplying the normalized intensities: $p_{i,cis} = I_{i,cargo} I_{i,cis}$ and $p_{i,trans} = I_{i,cargo} I_{i,trans}$ , where $I_{i,cargo}$, $I_{i,cis}$ , and $I_{i,trans}$ are the intensities of the cargo, and the cis-Golgi and trans-Golgi markers at the $i$-th pixel, normalized to the corresponding maximum intensity. The sums of all of the $p_{i,cis}$ and the $p_{i,trans}$ values represent the degree of co-localization of the cargo with the cis-Golgi and trans-Golgi markers, respectively. Using this method, the degree of overlap calculated between the Golgi markers themselves (GM130 and TGN46, which are in reality completely segregated in the cis-Golgi and the TGN, respectively) is about 30%. This means that the calculated distribution of GM130 itself between cis-Golgi and TGN is about 70% in the cis-Golgi and 30% in the TGN, and vice versa for TGN46 (see above). This method was also tested by comparing the GFP-albumin distribution in HeLa cells at steady-state as determined by this method, with that seen by immuno-EM. The degree of co-localization by method 2 of GFP-albumin with TGN46 was about 80% higher than with GM130 in good qualitative agreement with the immuno-EM data (*Figure 3—figure supplement 2*).

Finally, to visually verify the data obtained by Methods 1 and 2, we used a line-scan method. The intensity distributions of the fluorescently tagged proteins, as GM130 (red) and TGN46 (blue) and GFP-albumin (green), were plotted against distance. Importantly, all of these approaches gave results that were in good agreement under all of the conditions tested, and that also agreed well with the immuno-EM data.

## Electron microscopy and tomography

Cryo-immuno-EM, immuno-nanogold-labeling, rapid-freezing cryosubstitution, serial sectioning, 3D reconstructions, and electron tomography were all performed as previously described (*Polishchuk et al., 1999*; *Mironov et al., 2001*; *Trucco et al., 2004*). Correlative light-electron microscopy was performed as described previously (*Polishchuk et al., 2000*; *van Rijnsoever et al., 2008*; *Mironov and Beznoussenko, 2013*). All samples were analyzed under a Philips Tecnai-12 electron microscope (FEI/Philips Electron Optics, Eindhoven). High-pressure freezing of samples was carried out according to *Nicolas et al. (1989)*. For samples from rat liver, CD-COBS Charles River rats were anesthetized with Nembutal and sacrificed. Liver slices were fixed by immersion in 1% gluteraldehyde in 0.15 M HEPES (pH 7.2) for 1 hr and then post-fixed with 1% reduced $OsO_4$ for 2 hr on ice. For cultured cells, cryo-immuno-EM, cryosections of HepG2 cells were prepared and immunolabeled with antibodies against albumin, antitrypsin, GM130 and TGN46, and then analyzed as previously described (*Mironov et al., 2001*). Quantification of albumin, antitrypsin and VSVG labeling within the Golgi stacks was performed using the Analysis software (Soft Imaging Software Corporation; *Mironov and Mironov, 1998*). In studies of intra-Golgi transport, the number of gold particles were counted per cisterna in

30 stacks per time point (per experiment, in at least four experiments), and normalized to the value in the ER. This normalization was necessary to reduce the labeling variability between different sections and experiments. The TGN was defined as ribosome-less, tubular-reticular membranes adjacent to the trans-Golgi cisternae that were positive for TGN46 labeling. The IC elements were defined as clusters of tubular-vesicular membranes located near the *cis* side of a stack that were positive for ERGIC53. Vesicles were defined as round profiles adjacent (lateral) to Golgi cisternae and not exceeding 65 nm in diameter, a feature of COPI-derived vesicles. Golgi tubules were defined as elongated membrane profiles with length:width ratios of at least 1.5. For electron tomography, the analysis of intercisternal connections in chemically and cryo-fixed samples was performed on 200-nm-thick sections, passing roughly perpendicularly to the center of the Golgi stack, as described previously (*Trucco et al., 2004*). The numbers of intercisternal connections were counted in single tomograms and varied between 0 and 2 per tomogram. Because the average volume of each Golgi section represented approximately 20% of the total stack volume (conservatively assuming 'idealized' stacks made of round cisternae of about 1 micron in diameter), the number of connections detected in each tomogram was multiplied by five to obtain the total number of connections per stack. At least five tomograms were analyzed per experimental condition. Surfaces of Golgi membranes were rendered using the IMOD software (http://bio3d.colorado.edu/imod/).

## Photooxidation

For photooxidation experiments, HeLa cells transfected with GFP-albumin were fixed with a mixture of 4% formaldehyde and 0.05% gluteraldehyde for 5 min and then post-fixed with 2% formaldehyde for 20 min. Next, cells were washed and incubated with 0.1% DAB in 0.2 M cacodylate buffer (pH 7.4) on ice for 30 min. Then, cells immersed into DAB solution (still kept at 4°C) were placed under the LSM510 laser scanning confocal microscope (Zeiss) and an area of the Golgi complex exhibiting GFP fluorescence was irradiated with laser-derived blue (405 nm wave length) light with maximal intensity. After each scan of the light beam, there was at least a 10 s interval, allowing DAB to diffuse to the site of bleaching. When the fluorescent intensity of GFP emanating from the irradiated area had disappeared, we checked whether the expected brownish DAB precipitate was visible over the irradiated Golgi area. If a modest brownish staining was visible, cells were processed for EM using reduced $OsO_4$ and thiocarbohydrazide. Finally, 200 nm sections were subjected to electron tomography. For photo-oxidation experiments following FRAP studies, the Golgi region of the HeLa cells transfected with GFP tagged constructs was photobleached in the presence of culture medium and then after appropriate recovery time, the cells were fixed and subjected to photooxidation as above.

## Golgi vesiculation assay in vitro

Purified Golgi membranes were prepared from rat liver and incubated in vitro to induce the formation of vesicles, as described by Rothman et al. (*Malhotra et al., 1989*). All of the membranes were then pelleted and prepared for cryo-immuno-EM.

## Radioactive pulse-chase

The method was essentially as described (*Bonifacino, 2001*) with small modifications as described below. HepG2 cells infected with VSV were incubated at 32°C for 1h and then kept in methionine/cysteine free DMEM for 30 min. After which the media was substituted with one containing 0.2 mCi/ml of radiolabelled ($^{35}S$) cysteine and methionine for 5 min at 32°C and then incorporation of radioactivity was stopped by substituting with cold media. The cells were then incubated at 32°C for indicated times before lysing in RIPA buffer (150 mM NaCl, 20 mM Tris pH 8.0, 0.1% SDS, 0.5% Sodium deoxycholate and 1% Triton X-100) followed by immunoprecipitation with anti-antitrypsin and anti-VSVG antibodies for 6 hr at 4°C. The immunoprecipitate was then subjected to EndoH digestion before being resolved by SDS-PAGE followed by autoradiography.

## Endoglycosidase H treatment

The immunoprecipitates from the radioactive pulse-chase assay were eluted by incubating in the elution buffer (0.1 M sodium citrate pH 5.5, 0.5% SDS and 1% β-mercaptoethanol) at 90°C for 3–4 min followed by centrifugation at 13000×g for 5 min. The supernatant was then incubated with Endoglycosidase H (1000 units/ml) for 1h at 37°C. The treated samples were then resolved by SDS-PAGE followed by autoradiography.

## Computational modeling

All the simulations were performed using the MATLAB software together with the DIPimage toolbox (www.diplib.org; Hendriks CLL, Rieger B, van Ginkel M, van Kempen GMP, van Vliet LJ. DIPimage, a scientific image processing toolbox for MATLAB. Delft University of Technology, 1999.). See legends to *Figure 6*, *Figure 6—figure supplements 1 and 2* and the 3 Matlab scripts included in the supporting material for details of the individual simulations. The simulations of albumin diffusion through the Golgi stack were run both in 2D and 3D. Since the 2D and 3D concentration profiles were in very good agreement with the difference in equilibration time less than 12%, we only show data for diffusion in 2D. For the simulations, we used a mean field approach because we assume that the movement of the proteins in the experiments are mainly driven by diffusion. The diffusion equation $\partial c/\partial t = D \nabla^2 c$, with the albumin concentration $c$ and the diffusion coefficient $D$, was solved using the finite difference method with the *Forward Time, Centered Space* (FTCS) scheme (*Ames, 1992*). Both the time and space derivatives are replaced by finite differences

$c_{i,j}(t + \Delta t) = c_{i,j}(t) + D \, \Delta t/\Delta x^2 \, (c_{i-1,j}(t) + c_{i+1,j}(t) + c_{i,j-1}(t) + c_{i,j+1}(t) - n \, c_{i,j}(t))$,

where $c_{i,j}(t)$ is the concentration at the position $i$ and $j$ in x- and y-direction, respectively. The factor $n$ is the number of nearest neighbor pixels ($n = 4$ in the interior and less at the boundary). Space and time increments were chosen as 30 nm and 10 µs to ensure the stability criterion $D*\Delta t/\Delta x^2 < 1/2$. The change in the total particle number was monitored by measuring the integrated density (concentration) over time. The changes were between 1 and 4 particles out of about 20,000 particles over simulation times of 20–30 s. Dirichlet boundary conditions were applied, i.e. the albumin concentration was set to zero outside the cisternae and the tubules. The diffusion coefficient of albumin in the Golgi complex was taken as 10 µm²/s, which is a realistic value for a globular protein of 60 kDa in the cell cytosol (*Hobbie, 1978*; *Dauty and Verkman, 2004*). The system was simulated as a closed system with no flux from the ER and no flux out of the Golgi. Since the intra-Golgi diffusion rate of soluble cargoes is an order of magnitude faster than either entry into or exit out of the Golgi (*Figure 3—figure supplement 1*) excluding them in the simulations does not affect the conclusions.

Additionally the following assumptions are implicit in the model:

i. The protein is initially distributed homogeneously in the first cisternae.
ii. The initial protein density spreads out everywhere inside the confined volume (to other cisternae and tubules) according to the diffusion equation.
iii. The diffusion constant for albumin is homogenous throughout the system (inside the cisternae as well as inside the tubules).
iv. The protein concentration is high enough so that it can be described by density and not by stochastically moving individual proteins. This is not really a limitation because stochastic random walk motion yields the same diffusion behavior as a continuous particle density.
v. No interactions are considered between the diffusing proteins and the system boundaries (cisternae and tubule walls).

Of note, the use of modeling here was restricted to assess whether the equilibration rates of albumin across the stack are compatible with the continuity- or the vesicle-based model, or with both.

The size of the cisternae was varied and did not significantly change the magnitude of the time to reach equilibrium. Changing the diameter and length of the tubules affected the equilibration times in a way typical for diffusion processes (see figure legends for results). Also, systems with two tubules instead of one connecting adjacent cisternae were simulated. As mentioned above, three Matlab scripts are included in the supporting material that were used by us to produce the data for *Figure 6* and *Figure 6—figure supplement 1*.

## Statistics

All experiments involving immuno-EM and immunofluorescence were performed at least three times on different days, and each treatment was carried out as triplicate samples. For quantification, 15–30 individual measurements were made for each sample (for instance of the labeling density of albumin in Golgi stacks, or of the degree of co-localization of cargoes with a Golgi marker in a cell). For correlative light-electron microscopy (*Figure 3*), each experiment was carried out at least three times and at least three cells were examined. Experiments on the live dynamics of GFP-albumin (*Figure 3—figure supplement 2*) were repeated at least four times. Values are mean ±SD from

30 stacks per time point, in three independent experiments for immuno-EM; and mean ±SD of 10 co-localization measurements per time point for immunofluorescence.

## Acknowledgements

We thank D Corda and M A De Matteis for critical reading of the manuscript, DrVictor Hsu for critical contributions towards to the design of the study and experiments, C P Berrie for editorial assistance and R Le Donne for artwork preparation. We acknowledge the financial support of AIRC (Italian Association for Cancer Research, IG 10593), the MIUR Project "FaReBio di Qualità", the PON projects no. 01/00117 and 01-00862, PONa3-00025 (BIOforIU), PNR-CNR Aging Program 2012-2014 and Progetto Bandiera 'Epigen'. Research in the Helms group is supported by DFG via SFB 1027. RR was a recipient of FIRC Fellowship.

## Additional information

### Funding

| Funder | Author |
| --- | --- |
| Associazione Italiana per la Ricerca sul Cancro (AIRC) | Alberto Luini |
| Ministero dell'Istruzione, dell'Università e della Ricerca (Ministry of Education, Research and Universities) | Alberto Luini |
| Fondazione Italiana per al Ricerca sul Cancro (FIRC) | Alexandre A Mironov |

The funders had no role in study design, data collection and interpretation, or the decision to submit the work for publication.

### Author contributions

GVB, SP, AAM, Conception and design, Acquisition of data, Analysis and interpretation of data, Drafting or revising the article; RR, RP, OM, DDG, AF, MP, MRV, YGMR, Acquisition of data, Analysis and interpretation of data; AS, Acquisition of data, Analysis and interpretation of data, Drafting or revising the article; MS, MGC, Acquisition of data, Analysis and interpretation of data, Contributed unpublished essential data or reagents; VH, Conception and design, Acquisition of data, Analysis and interpretation of data; AL, Conception and design, Analysis and interpretation of data, Drafting or revising the article

## Additional files

### Supplementary files

• Supplementary file 1. Diffusion of albumin across the Golgi stack stable intercisternal tubules.

• Supplementary file 2. Diffusion of albumin across the Golgi stack via flickering intercisternal tubules.

• Supplementary file 3. Transport of albumin across the Golgi stack by vesicles.

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
