## [Decision Letter]

Thank you for sending your work entitled “Secretion of soluble proteins by diffusion via inter-cisternal continuities: a novel intra-Golgi transport mechanism” for consideration at *eLife*. Your article has been favorably evaluated by a Senior editor and 3 reviewers, one of whom is a member of our Board of Reviewing Editors. The Reviewing editor and the other reviewers discussed their comments before we reached this decision, and the Reviewing editor has assembled the following comments to help you prepare a revised submission.

A major controversy in the field of membrane traffic is how cargo progresses through the Golgi complex. Here the authors report a very important finding-namely that a soluble cargo and a large aggregate cargo (collagen) traverse a single Golgi stack at different rates. By itself, this finding would be inconsistent with simple cisternal maturation in that a means to traverse the Golgi without any cisternal movement is possible.

The work is carried out to a high standard but the manuscript needs reworking and certain parts need to be modified as described below. Figure S4 should really be a main figure combined with Figure 4, while the rest of Figure 4 could be moved to a new figure. The modelling sits at the end, and should be better integrated. One option would be to bring the simulation in earlier and explain what predictions the different models make. The data can then be presented in light of the simulation, commenting of which model provides the best fit to the experimental observations. Some of the EM is fantastic but other images could be a lot better. Detailed suggestions follow here:

1) The results show a correlation between the presence of cisternal continuities defined by EM, and cargo transport. They do not “demonstrate” cisternal continuities are responsible for transport as the authors claim in the text. For this conclusion the machinery mediating cisternal continuity formation would need to be inactivated, and further experiments done showing (i) cisternal continuities are lost and (ii) transport ceases. The major question left unresolved by this work is the mechanism by which cisternal continuities between adjacent Golgi compartments form and resolve. A further question is how this occurs without collapse and mixing of the two compartments to generate one larger compartment. Please acknowledge and discuss.

2) In the Results section, the authors describe albumin transport following release of a 15C block. They refer to Figure 1-m, however this fails to convincingly show accumulation of albumin in the Golgi region marked by either GM130 or TGN46. The diffuse staining, possibly ER as the authors mention, does not seem to be reduced and accounts for the bulk of the signal while only a small proportion overlaps with Golgi markers. This suggests that albumin trafficking is not particularly efficient and only a subpopulation is being transported to the Golgi. These conditions are used for the immunoelectron microscopy, so this is a problem for all the data shown. The reviewers would prefer to see more examples of the triple labeling shown in Figure 1 than that shown in j-q.

3) Please remove the Concanamycin data (and mention of these data) in Figure S2 – they don't add to the story.

4) In Figure 2 albumin is compared to procollagen (PC) traffic. In panel 2a a whole cell is shown, while for the other condition (3min at 32C) only a magnified region of the Golgi is shown. Figure 2 showing PC localization by EM is described in the text before Figure 2 showing albumin. The figure order should be changed. A more serious problem is the quality of the PC image in Figure 2 does not provide strong support that PC is in the “cis but not the distal” Golgi cisternae as claimed in the manuscript. Was a Golgi marker used to define the cis cisternae? If yes, then a clearer image with highlights for the different markers as nicely done in Figure 1 should be shown.

The data in Figure 3 are also far from compelling evidence that albumin rapidly fills the entire Golgi stack while PC moves more slowly. It is not obvious to me what the CLEM data in Figure 3o-r show, and this is not quantitated. The images in Figure 3 are not shown at the same scale. This isn't entirely clear and the figure legend is confusing - it would be better if the scale was marked in the figure and not appended to the legend. Panels 3a-c show whole cells, while 3d-i are Golgi only however at different scales. It would be better if cells were shown at one magnification while enlarged Golgi was shown at another standardized scale to allow simpler comparison of the different images and between figures.

5) The cisternal continuity data in Figure 4 appears to rely on arbitrary tracing of membrane edges. The example in Figure 4 is not an unambiguous connection and the yellow labeled cisternae in Figure 4 appears to have a clear membrane bilayer in the region said to form a join. I have a similar problem with Figure 4, apparently showing a continuity labeled for albumin that lies adjacent to a Golgi stack but isn't definitely connected to it.

6) The model in Figure 5 is presented to give a framework for the continuity model for trafficking. This isn't well integrated with the rest of the work. The assumptions are not properly described in the text, and exactly what testable predictions the modeling generates is unclear. Why does VSV G not diffuse through cisternal continuities as rapidly as soluble cargo? From what I can tell the modeling is a simplification of the 3D membrane architecture of the Golgi to 2D, so surface area to volume ratio is not correctly accounted for. What are the absolute concentrations of albumin, VSV G and PC in the system? My concern is that albumin levels are much higher than the other markers, and this may explain some of the differences seen.

7) Two minutes is very fast for diffusion through continuities. Have the authors considered the issue of whether fixation (or cryoEM processing) is fast enough to have full confidence in the 2 minute time point (versus 5 minutes, for example)? It does not change their story but is an important issue to control for experimentally.

8) The state distribution of albumin in HepG2 cells seems more complicated than their simple model. If the compartment can fill in 2 minutes and the ER has little albumin, why does the compartment empty so much more slowly? Addition of cycloheximide and determination of the efflux rate would be informative here. How can TGN accumulation be explained? Is the albumin binding to lipidic particles in the Golgi to slow its exit? Please comment on this in relation to use of vesicles versus simple diffusion.

9) The authors report that small diffusive cargos cross the Golgi complex in a few minutes. It may be hard to reconcile this view with the multi-step glycosylation at work for certain cargos. It would be important to know whether glycosylation kinetics fits with the observed transport kinetics. The alpha1-antitrypsin reporter chosen in this study may be an adequate example. The use of lectin may be helpful to carry out a rough analysis.

10) The diffusive model explains very well how a cargo may fill-up every cisternae in a very short time. However, it seems to me that the export will then be very slow if no additional hypothesis is used to impose that the diffusion is biased in the anterograde direction. Otherwise, the cargo will always be homogeneously distributed in all cisternae and emptying the Golgi would be very slow. The authors mention pH gradient but without really using this as driving force. In addition, not all cargo may be sensitive to the pH gradient that exists in the Golgi. The authors should take this in account in their simulations and try to estimate the exit time (e.g. time need to empty 90% of the Golgi) of small and large cargo in the absence of biased diffusion. They should discuss this point.

11) The authors may measure the exit time of the small cargo (in an experiment similar to the one presented in Figure S3b) and compare the exit of large cargo exiting Golgi through maturation. These values may be used in the simulation to see what level of diffusion bias one has to add to the diffusion model to explain the observed kinetics.

12) While the FRAP experiment is showing a fast filling of the Golgi, bleaching half, and not all, the Golgi would also have giving interesting diffusion value. Ideally, the bleaching of all the ER should be done at the same time to separate ER-Golgi and intra-Golgi recovery. It may also enable to detect differences of diffusion in Golgi sub-domains.

13) The use of 15degree block and release may induce differences between small cargos like albumin and large ones (like VSVG) that may not be visible in normal conditions. It is clear that retention at 15 degree is already not occurring in the same sub-compartment for the different cargos and this difference may explain part of the downstream behaviour reported here. Carrying out a photo-bleaching experiment to compare the kinetics of intra Golgi diffusion of small and large cargos may help to control for this in physiological conditions.

---

## [Author Response]

*The work is carried out to a high standard but the manuscript needs reworking and certain parts need to be modified as described below*.

We thank the reviewers for the appreciation of our work and their constructive criticism.

*Figure S4 should really be a main figure combined with*
Figure 4*, while the rest of*
Figure 4
*could be moved to a new figure*.

We have done this and modified the manuscript appropriately (see new Figure 4).

*The modelling sits at the end, and should be better integrated. One option would be to bring the simulation in earlier and explain what predictions the different models make. The data can then be presented in light of the simulation, commenting of which model provides the best fit to the experimental observations*.

We have sought to modify the manuscript to take into account the perception of the reviewers that the models should be better integrated. In order to achieve this, we need to consider that the scope of our models is limited. The models only aim to assess the equilibration rates of small soluble cargoes across a closed system with a stack-like geometry, and in particular to compare two possibilities: a) cargo equilibration through the stack by diffusion via continuities and b) equilibration via shuttling vesicles. Importantly, the models do not aim to simulate the entire traffic process through the Golgi including cargo arrival, departure and intra-Golgi concentration steps. Given these limitations, we feel that it would be difficult for a reader to grasp the significance of these models without first providing the appropriate experimental background. Rather, we think that the best way to use the models in this manuscript is to go through the following logical steps:

- The experiments describe the morphology of the stacks including tubular connections between cisternae, suggesting that these connections are used to translocate soluble proteins like albumin. The experiments also establish an upper bound of 2 minutes, as the time required for albumin passage from the first to the last compartment.

- These data pose the question as to whether the equilibration of albumin through the stack within 1-2 min can be best explained by a model based on simple diffusion of albumin between the cisternae through tubules without biasing forces, or by a model based on vesicular transport.

- The calculations reveal that the diffusion-based model is easily compatible with the experimental data, while the vesicular model is difficult (or impossible) to reconcile with the data, since a great number of vesicles would be required to explain the observed transport rates whereas no evidence is available from the experimental side that such a high amount of vesicle traffic exists.

We therefore propose to improve on the original scheme, rather than to drastically change it. In the revised version, we seek to achieve a better integration of the models by making reference to the original data on which the parameters of the models were based on, and by clarifying the logical flow that was adopted. However, if the editor and the reviewers strongly prefer to stick to their original advice to restructure the manuscript and bring the simulation in earlier, we will find a way to do so.

*Some of the EM is fantastic but other images could be a lot better*. *Detailed suggestions follow here:*

*1) The results show a correlation between the presence of cisternal continuities defined by EM, and cargo transport. They do not “demonstrate” cisternal continuities are responsible for transport as the authors claim in the text. For this conclusion the machinery mediating cisternal continuity formation would need to be inactivated, and further experiments done showing (i) cisternal continuities are lost and (ii) transport ceases*.

We agree with the reviewers and we have appropriately changed the word from “demonstrate” to “consistent with”. The sentence now reads as follows: “Thus, these results provide evidence consistent with a novel mechanism of transport for a major class of secretory proteins, and for the notion of multiplicity of transport mechanisms that can help to rationalize most of the observed intra-Golgi trafficking patterns”.

*The major question left unresolved by this work is the mechanism by which cisternal continuities between adjacent Golgi compartments form and resolve*.

We agree with comment of the reviewers that the mechanistic question is not fully resolved, although a certain amount of information is available on the mechanism of formation and fission of the intercisternal tubules. We have now added a note regarding this point in the Introduction (“At the molecular/mechanistic level … a complete understanding of the molecular players regulating the intra-Golgi connections remains lacking”).

*A further question is how this occurs without collapse and mixing of the two compartments to generate one larger compartment. Please acknowledge and discuss*.

The issue of coexistence of intercisternal connections and cis-trans polarity of the Golgi apparatus is not without antecedents. We have discussed various possibilities at length and now suitably modified the Discussion to highlight this point (“The second question raised by the cargo diffusion model is how the Golgi maintains its compositional polarity … Clearly, much work is needed to clarify whether and how some of these mechanisms apply”).

*2) In the Results section, the authors describe albumin transport following release of a 15C block. They refer to*
Figure 1*, however this fails to convincingly show accumulation of albumin in the Golgi region marked by either GM130 or TGN46. The diffuse staining, possibly ER as the authors mention, does not seem to be reduced and accounts for the bulk of the signal while only a small proportion overlaps with Golgi markers. This suggests that albumin trafficking is not particularly efficient and only a subpopulation is being transported to the Golgi. These conditions are used for the immunoelectron microscopy, so this is a problem for all the data shown. The reviewers would prefer to see more examples of the triple labeling shown in*
Figure 1
*than that shown in j-q*.

As suggested by the reviewers we have now included more images of cryo-immunogold microcopy that define the distribution of albumin across the Golgi at high resolution (see new Figure 1).

Regarding the time points shown in Figure 1-m (now Figure 1—figure supplement 1), we note that they are the earliest time points – 2min after release of 15°C block. The rest of the time course is presented in Figure 1—figure supplement 1 where one can see that by 5 min the ER has already emptied to a large extent, indicating that the transport of albumin is remarkably efficient. Moreover, we note that even at 2 min after release of the 15°C block, although albumin is still mostly in the ER, it has also already reached the Golgi, as indicated by the presence of albumin fluorescence within the Golgi area (to be compared with the absence of albumin fluorescence at time 0). Thus, under the conditions used for the experiments, the albumin transport appears to be efficient. We have clarified the text in this regard. The text reads as follows: “Albumin showed a diffuse ER-like distribution at time 0, with no clear Golgi staining (Figure 1—figure supplement 1); … the export of albumin out of the ER was very efficient, so that by 10 min after the release of the temperature block almost all of the protein had been transported to the Golgi apparatus.”

*3) Please remove the Concanamycin data (and mention of these data) in Figure S2 – they don't add to the story*.

We have removed the Concanamycin data as suggested by the reviewer. Since one of the latter questions (see point 10) raised by the reviewers directly refers to the role of pH in the albumin transport we had to retain the reference to this data and we have mentioned it as data not shown.

*4) In*
Figure 2
*albumin is compared to procollagen (PC) traffic. In panel 2a a whole cell is shown, while for the other condition (3min at 32C) only a magnified region of the Golgi is shown.*
Figure 2
*showing PC localization by EM is described in the text before*
Figure 2
*showing albumin. The figure order should be changed*.

We have changed the images in Figure 2, which now show enlarged Golgi in all conditions, so that they are now uniform. We have also changed the order of the figures so that PC localization by EM is presented before that of albumin.

*A more serious problem is the quality of the PC image in*
Figure 2
*does not provide strong support that PC is in the “cis but not the distal” Golgi cisternae as claimed in the manuscript. Was a Golgi marker used to define the cis cisternae? If yes, then a clearer image with highlights for the different markers as nicely done in*
Figure 1
*should be shown*.

We apologize for not being clear with the image. A Golgi marker was indeed used to define the cis-Golgi in Figure 2. While PC can be identified morphologically by the presence of aggregates (now indicated by an asterisk), the cis-Golgi was identified by using GM130 as a marker (black dots) labeled by immuno nanogold technique. The gold particles showing the presence of GM130 are now indicated by arrows. We have not done cryo-immunogold labeling (usually used to label two or three markers simultaneously) here, since PC can easily be identified morphologically and only single labeling for the cis-Golgi marker was required.

*The data in*
Figure 3
*are also far from compelling evidence that albumin rapidly fills the entire Golgi stack while PC moves more slowly. It is not obvious to me what the CLEM data in Figure 3o-r show, and this is not quantitated*.

The experiments in Figure 3 complete the experiments described in Figures 1 and 2, where we show that under synchronized conditions albumin traverses the Golgi stack faster than VSVG or PC. In other words, the experiments in Figure 3 are meant to examine whether the observed fast kinetics of albumin transport might be limited to the synchronization conditions (Figures 1 and 2) or they are observed also at steady state. The results show that, under steady state conditions, the newly recovered fluorescence of albumin (immediately after photobleaching), that represents newly arrived cargo from the ER at the Golgi apparatus, can be seen to spread throughout the Golgi (as determined from the overlap of albumin with both GM130 and TGN46), while in the case of PC, it is mainly restricted to the cis side of the Golgi. We think that such an overlap is fairly compelling evidence that the albumin traverses the Golgi stack faster than PC even under steady state conditions. Moreover the use of Golgi ministacks where the separation between the cis and trans Golgi is always clearer than that seen with the intact Golgi ribbon provides a further convincing argument that even under steady state conditions the albumin traverses the Golgi stack faster than VSVG or PC.

*It is not obvious to me what the CLEM data in*
Figure 3
*show, and this is not quantitated*.

In addition, to confirm the above light microscopy data with a high resolution EM-based approach, we used a method based on photo-oxidation of GFP coupled to FRAP to unambiguously identify the newly arrived cargo, represented by the photoconversion product (marked by arrows in Figure 3). As can be seen from the figure in the case of albumin the photo-conversion product is present throughout the Golgi already at 2 min after bleaching, suggesting rapid transport across the stack (consistent with all of the previous data), while it is restricted to the cis side in the case of VSVG and PC, which migrate through Golgi more slowly.

Regarding quantification, due to intrinsic limitations of the technique, the density of the photo-conversion product cannot be quantified in a meaningful way. However, as requested by the reviewer, we have now added quantifications by measuring the percentage of cells where the Golgi was completely filled with GFP-albumin and cells where GFP-albumin is present only at the cis side (see Figure 3).

We apologize for not having been clear about the description of the experiments represented in Figure 3. We have rewritten the corresponding figure legend to make it clearer.

*The images in*
Figure 3
*are not shown at the same scale. This isn't entirely clear and the figure legend is confusing – it would be better if the scale was marked in the figure and not appended to the legend. Panels 3a-c show whole cells, while 3d-i are Golgi only however at different scales. It would be better if cells were shown at one magnification while enlarged Golgi was shown at another standardized scale to allow simpler comparison of the different images and between figures*.

We apologize for the confusion. We have now changed the scale bars in this figure so that the bars represent a uniform length (2μM) to make the comparison between different images easier. We have not marked the scale in this figure to maintain uniformity with other figures.

*5) The cisternal continuity data in*
Figure 4
*appears to rely on arbitrary tracing of membrane edges. The example in*
Figure 4
*is not an unambiguous connection and the yellow labeled cisternae in*
Figure 4
*appears to have a clear membrane bilayer in the region said to form a join. I have a similar problem with*
Figure 4*, apparently showing a continuity labeled for albumin that lies adjacent to a Golgi stack but isn't definitely connected to it*.

We again apologize for not being clear with the image in Figure 4. The image is one virtual section from the tomogram shown in Figure S4 (now Figure 4). When seen in the context of the whole panel of successive images the reviewer can clearly see that the intercisternal connection is a valid one. To avoid any confusion, we have removed panel 4c and shifted Figure S4 to the main figures. The new figure (Figure 4) contains the whole set of virtual sections from the tomogram making it easier to visualize the rather tortuous intercisternal connection. We also note that most connections are fairly complex and require an analysis of several successive virtual sections, which explains why they are rarely detected by casual observers.

Regarding Figure 4 (now Figure. 5f), this is one of the examples of ‘simple’ or ‘linear’ connections between two distinct cisternae in the Golgi stack. As discussed in the text, these linear continuities are rare in thin sections. We chose this one because the labeling is particularly ‘clean’, and because the connection can be discerned, especially on the ‘internal’ side of the turn of the tubule. We have now added arrows to facilitate the appreciation of the continuity. Following the arrows one can clearly see the luminal continuity between the two non-adjacent cisternae that the tubule connects.

*6) The model in*
Figure 5
*is presented to give a framework for the continuity model for trafficking. This isn't well integrated with the rest of the work. The assumptions are not properly described in the text, and exactly what testable predictions the modeling generates is unclear*.

Part of our response to these comments can be found above. Basically, the two models, one based on equilibration via continuities and one on equilibration via shuttling vesicles, are simple, and are designed only to assess whether the equilibration rates of albumin across the stack are compatible with a continuity-based, or with a vesicle–based mechanism, or with both. Here, we will consider the above comments one by one.

The assumptions are not properly described in the text…

The parameters that were used for the modeling were not described in detail in the main text to enhance the readability of the manuscript. Nevertheless, detailed descriptions of the necessary parameters are in the figure legends, and are now clearly referred to in the main text, and the scripts used for the modeling are also provided as supplementary material. Additionally, the technical aspects of the modeling and the assumptions made are now described in detail in the Methods section. The assumptions of the model are recapitulated briefly below:

i) The protein is distributed homogenously in the first cisternae.

ii) The initial protein density spreads out everywhere inside the confined volume (to other cisternae and tubules) according to the diffusion equation.

iii) The diffusion constant for albumin is homogenous throughout the system (inside the cisternae as well as inside the tubules).

iv) The protein concentration is high enough so that it can be described by density and not by stochastically moving individual proteins. This is not really a limitation because stochastic random walk motion yields the same diffusion behavior as a continuous particle density.

v) No interactions are considered between the diffusing proteins and the system boundaries (cisternae and tubule walls).

*This isn't well integrated with the rest of the work*.

As detailed above, the use of modeling is restricted to testing which of the two models (diffusion based or vesicle transport based) can be reconciled with the experimental data. To this end we judge that a reader will be unable to grasp the significance of these models without first providing the appropriate experimental background. Thus we have followed the logical progression of first describing the experimental data (an upper bound of 2 minutes for albumin equilibration through the stack), then building computational models of the two competing hypothesis (transport based on diffusion of cargo or transport mediated by vesicles) and finally concluding that only the diffusion based model is compatible with the experimental data.

In the revised version, in order to better integrate the models, while describing them we make reference to the original data on which the parameters of the models are based on, and also clarify the logical flow that was adopted.

*…exactly what testable predictions the modeling generates is unclear*.

The diffusion-based model and the vesicle shuttling model aim to calculate the time required for albumin equilibration across the stack and the number of events (vesicle budding/fusion) required for equilibration, respectively. For the purpose of this study, the models aim only to assess whether the equilibration rates of albumin across the stack are compatible with the continuity– or the vesicle–based model, or with both.

In the revised version, we have clarified the limitations of the models and the logical flow that was adopted.

We have also added a note in the Methods section that reads as follows: “Of note, the use of modeling here was restricted to assess whether the equilibration rates of albumin across the stack are compatible with the continuity, or the vesicle-based model, or with both.”

Why does VSV G not diffuse through cisternal continuities as rapidly as soluble cargo?

Regarding VSVG, information that is not often appreciated but which emerges from the literature, and that has been briefly discussed in our manuscript, is that VSVG can behave in two different modes: it can equilibrate rapidly through the stack like albumin or it can behave like PC, depending on the synchronization conditions. Specifically, the albumin-like behavior of VSVG has been reported to occur (5; 59) when VSVG is accumulated in the ER at 40°C and then released at 32°C. Instead, under the conditions we use for synchronization (40-15-32°C), VSVG exhibits a transport behavior similar to that of PC, possibly because it forms oligomers/polymers (29) that behave like PC during the 15°C block (50; 84). We have modified and extended these comments in the Discussion section.

*From what I can tell the modeling is a simplification of the 3D membrane architecture of the Golgi to 2D, so surface area to volume ratio is not correctly accounted for*.

We agree with the reviewers that diffusion in 3D generally differs from diffusion in 2D. For diffusion in 3D, the mean squared displacement <*x*^2^> during a time interval *t* is <*x*^2^> = *6 D t* with D being the diffusion coefficient, whereas for diffusion in 2D, <*x*^2^> = *4 D t*. In the present situation, however, the 2D assumption perfectly matches the biological system.

Nevertheless, we have also compared 2D and 3D diffusion of albumin particles through the idealized Golgi compartment shown in Figure 5 (now Figure 6) with the simulation package Virtual Cell (http://www.nrcam.uchc.edu/). We obtained the following results for a standard configuration with 6 cisternae, with a cisternae thickness of 20 nm, cisternae diameter of 1000 nm, tubule length of 100nm, tubule diameter of 30 nm, and diffusion coefficient D of 10 μm^2^/s:

- assuming diffusion in 3D, 50% equilibration in compartment #6 was reached after 2.3 s and 95 % equilibration after 6.7 s.

- assuming diffusion in 2D, 50 % equilibration in compartment #6 was reached after 2.6 s and 95 % equilibration after 7.5 s.

We have additionally performed the same comparison for a system with large cisternae (cisternae diameter of 1500 nm) and keeping the other parameters same as before:

- assuming diffusion in 3D, 50 % equilibration in compartment #6 was reached after 5.4 s and 95 % equilibration after 15.7 s.

- assuming diffusion in 2D, 50 % equilibration in compartment #6 was reached after 6.0 s and 95 % equilibration after 17.3 s.

The results for the two system sizes are therefore very similar. In all cases, the difference in the equilibration time is less than 12%, showing that simulating in 2D leads to the same behavior as in 3D. We believe this justifies our simplification to the 2D case as used in the manuscript.

This is now mentioned in the Methods section of the revised manuscript. We now state:

“The simulations of albumin diffusion through the Golgi stack were run both in 2D and 3D. Since the 2D and 3D concentration profiles were in very good agreement with the difference in equilibration time less than 12%, we only show data for diffusion in 2D.”

*What are the absolute concentrations of albumin, VSV G and PC in the system? My concern is that albumin levels are much higher than the other markers, and this may explain some of the differences seen*.

To address this point, we refer the reviewers to the new Figure 3—figure supplement 2 where we now show biochemical pulse-chase analysis of antitrypsin and VSVG in HepG2 cells under conditions similar to that used in Figure 1 (transport assay monitored by EM). As noted in the text, the transport kinetics of antitrypsin are indistinguishable from those of albumin. The conditions of immunoprecipitation were such that the proteins were completely depleted. One can notice that the amounts of antitrypsin and VSVG are similar (in terms of radioactive counts or here intensity of the protein bands), suggesting that the antitrypsin is not present in much higher levels than VSVG. Therefore, the difference between their kinetic behaviors is not due to the different abundance of VSVG and antitrypsin and, by extension, of albumin. We have now included the following phrase in the legend to Figure 3—figure supplement 2: it is important to note here that the quantities of antitrypsin and VSVG present are very similar suggesting that the difference in the transport behavior of these proteins is not due to differences in their abundance. In addition, as mentioned in the text, the transport behaviors of antitrypsin and albumin are similar, reiterating further that the differences in the transport behavior between soluble secretory cargoes (albumin and antitrypsin) and VSVG/PC is possibly not due to the differences in their abundance.

*7) Two minutes is very fast for diffusion through continuities. Have the authors considered the issue of whether fixation (or cryoEM processing) is fast enough to have full confidence in the 2 minute time point (versus 5 minutes, for example)? It does not change their story but is an important issue to control for experimentally*.

In our experience, 2 min is ample time to fix the samples. This was addressed in our own original paper on the development of correlative video-electron microscopy (65). The data indicated that fixation of the cell in monolayers takes only 1-2 sec. Published literature in this aspect has shown that the fixation happens in a matter of seconds. The literature in this regard has been reviewed extensively in “Fine structure Immunocytochemistry“ by Gareth Griffiths (Griffiths, 1993). This aspect is now briefly mentioned in the Methods section. It reads as follows: “It is important to note here that fixation of monolayers of cells is a rapid process that happens in a time frame of 1-2 sec (65), and so any artifact due to slow fixation of the samples is nonexistent.”

Points 8-11

Since some of the issues raised by the reviewers are to some extent redundant, we prefer to group our responses as detailed below, for the sake of clarity and brevity.

The reviewers raise a series of interesting questions on the relationship between the diffusion-based mechanisms of intra-Golgi traffic and the mechanism of export from the Golgi. We agree that these questions are interesting and important in their own right; however we respectfully note that this study is meant to focus on intra-Golgi traffic and that an analysis of the mechanism and rate of exit of soluble cargoes from the Golgi is beyond its scope.

Nevertheless, we have carried out some of the experiments and discussed the relevant aspects mentioned by the reviewers in the revised manuscript along the lines reported below: “Another point of consideration while discussing the cargo diffusion model is the possible relationship between intra-Golgi transport and Golgi export […] Thus, the fact that export is pH-dependent is a further indication in favor of the diffusion-based mechanism.”

*12) While the FRAP experiment is showing a fast filling of the Golgi, bleaching half, and not all, the Golgi would also have giving interesting diffusion value. Ideally, the bleaching of all the ER should be done at the same time to separate ER-Golgi and intra-Golgi recovery. It may also enable to detect differences of diffusion in Golgi sub-domains*.

We are not sure we understand the comments of the reviewers. In any case, the bleaching of half of the Golgi and assessment of the ‘longitudinal’ diffusion rate has been performed (Figure 3—figure supplement 1). The half-time of ER-Golgi transport (200sec) and the intra-Golgi recovery (20sec) differ by an order of magnitude and so the contribution of ER-Golgi transport to the intra-Golgi recovery would be minimal. Moreover, these data were used merely to show that the GFP-albumin behaves as expected and the quantitation does not bear directly on the main conclusions of the manuscript. We have now mentioned this in the Methods section. It reads as follows:

“The same approach was used to measure the diffusion of albumin along the Golgi ribbon. Since the rate of diffusion of albumin along the Golgi ribbon was an order of magnitude faster than the ER to Golgi transport of albumin (Figure 3—figure supplement 1) the contribution of the latter to the recovery should be negligible. For exit from the Golgi complex, the FRAP-related iFRAP technique was used (59).”

We also agree with the reviewers that studying the differences, if any, in the diffusion kinetics in Golgi sub-domains would be interesting. However, our opinion is that the results of such a study would not affect the conclusions reached in the manuscript; rather they present a stimulating starting point for a follow-up to the present study.

*13) The use of 15degree block and release may induce differences between small cargos like albumin and large ones (like VSVG) that may not be visible in normal conditions. It is clear that retention at 15 degree is already not occurring in the same sub-compartment for the different cargos and this difference may explain part of the downstream behaviour reported here. Carrying out a photo-bleaching experiment to compare the kinetics of intra Golgi diffusion of small and large cargos may help to control for this in physiological conditions*.

We have studied the intra-Golgi diffusion of GFP-albumin, which appears to be much faster than the intra-Golgi diffusion of VSVG-GFP (59) or PCIII-GFP (which does not diffuse; our unpublished observations), thus indicating that albumin can indeed diffuse rapidly ‘horizontally’ as well as ‘vertically’ across the Golgi stacks, as expected form its soluble nature and the presence of connections.

The reviewers note that at 15C albumin resides both in the IC and the ER, while VSVG concentrate in the IC, and that this might explain their different downstream behaviors. It is difficult to see how these different distributions of the two cargoes might lead to a much more rapid filling of the Golgi by albumin than by VSVG. In any case, it is to control for this and other possibilities that we have studied the kinetics of transport at steady-state with no synchronization (see Figure 3). Also under steady-state conditions albumin still shows a faster kinetics of intra-Golgi transport compared to VSVG and other cargoes, further reinforcing our conclusion that the transport behavior of small soluble cargoes is different from others.